

# Spatial and seasonal variations of aerosols over China from two decades of multi-satellite observations. Part I: ATSR (1995-2011) and MODIS C6.1 (2000-2017)

Larisa Sogacheva[1*], Gerrit de Leeuw[1], Edith Rodriguez[1], Pekka Kolmonen[1], Aristeidis K. Georgoulias[2], Georgia Alexandri[2], Konstantinos Kourtidis[2], Emmanouil Proestakis[3,4], Eleni Marinou[5], Vassilis Amiridis[3],Yong Xue[6], Ronald J. van der A[7]

[1]Finnish Meteorological Institute (FMI), Climate Research Programme, Helsinki, Finland
[2]Laboratory of Atmospheric Pollution and Pollution Control Engineering of Atmospheric Pollutants, Department of Environmental Engineering, Democritus University of Thrace, Xanthi, Greece
[3] National Observatory Athens (NOA), Greece
[4]Laboratory of Atmospheric Physics, Department of Physics, University of Patras, 26500, Greece
[5]Deutsches Zentrum für Luft und Raumfahrt (DLR), Institut für Physik der Atmosphäre, Oberpfaffenhofen, Germany
[6]Department of Electronics, Computing and Mathematics, College of Engineering and Technology, University of Derby, Derby DE22 1GB, UK
[7]Royal Netherlands Meteorological Institute (KNMI), De Bilt, Netherlands

* Correspondence to: Larisa Sogacheva (larisa.sogacheva@fmi.fi)

**Abstract.**

Aerosol optical depth (AOD) patterns and interannual and seasonal variations over China are discussed based on the AOD retrieved from the Along-Track Scanning Radiometer (ATSR-2, 1995-2002), the Advanced ATSR (AATSR, 2002-2012) (together ATSR) and the Moderate Resolution Imaging spectrometer (MODIS) aboard the Terra satellite (2000-2017). The AOD products used were the ATSR Dual View (ADV) v2.31 AOD and MODIS/Terra Collection 6.1 (C6.1) merged dark target (DT) and deep blue (DB) AOD product. Together these data sets provide an AOD time series for 23 years, from 1995 to 2017.

Differences between the MODIS C6.1 and C6 AOD products with respect to AOD coverage and validation with Aerosol Robotic Network (AERONET) over China are briefly discussed, showing better validation results for C6.1: the correlation coefficient has increased from 0.9 in C6 to 0.92 in C6.1; bias has been slightly lowered from 0.007 to 0.006.

ADV and MODIS AOD validation results show similar high correlation with AERONET AOD (0.88 and 0.92, respectively), while the corresponding bias is positive for MODIS (0.06) and negative for ADV (-0.07). Validation of the AOD products in similar conditions, when ATSR and MODIS/Terra



overpasses are within 90 minutes from each other and when both ADV and MODIS retrieve AOD, show that ADV performs better than MODIS in autumn, while MODIS performs slightly better in spring and summer. In winter, both ADV and MODIS underestimate AERONET AOD.

Similar AOD patterns are observed by ADV and MODIS in annual and seasonal aggregates. ADV-MODIS difference maps show that MODIS AOD is generally higher than that from ADV.

During the period 1995-2006, AOD was increasing in the SE of China. Between 2006 and 2011, AOD was not changing much, showing minor minima in 2008-2009. From 2011 onward, AOD is decreasing in the SE of China.

Both ADV and MODIS show similar seasonal AOD behavior. The spring AOD maxima in the south is shifting from spring to summer along the eastern coast in the direction to the north.

Similar patterns exist in year-to-year ADV and MODIS annual AOD tendencies in the overlapping period. The agreement between sensors in year-to-
year AOD changes is quite good. The large AOD differences exist between regions, for both sensors.

The consistency between ATSR and MODIS as regards the AOD tendencies in the overlapping period is rather strong in summer, autumn and overall for the yearly average, while in winter and spring, when there is a difference in coverage between the two instruments, the agreement in the AOD tendencies is lower.

## 1 Introduction

The concentrations of aerosols in China have been relatively high since over two decades (e.g., Wang et al., 2017; Zhang et al., 2017) and rising as a consequence of economic development, industrialization, urbanization and associated transport and traffic. Another factors exist which cause the interannual and seasonal variations in AOD over China, such as the generation and transport of desert dust (e.g. Proestakis et al., 2018, Wang et al., 2008) seasonal biomass burning (e.g., Chen J. et al., 2017) as well as meteorological conditions and large-scale circulation (Zhu et al., 2012). Both direct production of aerosol particles and the emission of aerosol precursor gases, such as $SO_2$, $NO_2$ and volatile organic compounds (VOCs), contribute to the
observed aerosol concentrations which manifest themselves as particulate matter (PM) or aerosol optical depth (AOD) (Bouarar et al., 2017). PM2.5, the dry mass of aerosol particles with an ambient diameter smaller than 2.5 µm, is often used in air quality and health studies as a measure for aerosol concentrations. PM2.5 is a local quantity that is usually measured near the surface. In contrast, AOD is the column-integrated extinction coefficient, which is an optical property commonly used in climate studies, and can be measured from satellites or ground-based remote sensing. PM2.5 and AOD, although both used as a measure for the occurrence of aerosols, are very different aerosol properties which cannot be directly compared.

Spatial variation, seasonal variability and time series have been observed from the analysis of ground-based networks measuring aerosol optical properties using sun photometers in CARSNET (Che et al., 2015), hand-held sun photometers in the Chinese Sun Hazemeter (CSHNET; Wang, Y., et al., 2011), CARE-China (Xin et al., 2015) and SONET (Li et al., 2017). These networks provide point measurements, which are representative for a limited area,



and the coverage across China still leaves large gaps. Satellite observations, although less accurate, fill these gaps and provide a clear picture of spatial and temporal variations across the whole country.

In de Leeuw et al. (2018), was shown how the complementary use of three radiometers, the Along-Track Scanning Radiometer ATSR-2 on ERS-2, the Advanced ATSR (AATSR) aboard the environmental satellite Envisat, and the MODerate resolution Imaging Spectroradiometer (MODIS) on Terra

results in two decades (1995-2015) of AOD observations over mainland China. This information was complemented with observations from the Cloud Aerosol Lidar with Orthogonal Polarization (CALIOP) between 01/2007 and 12/2015 on the aerosol vertical structure. The satellite data show the high aerosol concentrations over distinct regions in China such as the North China Plain (NCP) including the Beijing-Tianjin-Hebei (BTH) area, the Yangtze River Delta (YRD), the Pearl River Delta (PRD) and Sichuan province/Chongqing, as well as the enhanced AOD over the Taklamakan desert (TD).

The two-decadal time series show the initial rise of the aerosol burden over China at the end of the 1990s followed by AOD variations in response to

policy measures to improve air quality by the reduction of emissions of both aerosol particles and their precursor gases such as $SO_2$ and $NO_2$ (e.g. van der A et al., 2017). After 2011, the AOD appears to decrease toward the end of the study period used in de Leeuw et al. (2018), i.e. the end of 2015. These observed temporal variations of the AOD have also been reported elsewhere, including recent analyses indicating the decline since about 2011 (Zhang et al., 2017; Zhao et al, 2017) with some variation in the reported pivot point. An interesting question whether the recent decrease in AOD is confirmed by extension of the time series with the most recent data is addressed in the current paper and studied more detailed in Sogacheva et al., 2018, submitted to ACPD, hereunder referred as Part II.

Most of the aerosol studies over China are focused on the SE part of the country or on specific regions or cities in SE China. However, the economic situation and Governmental policy measures to improve air quality by emission reduction obviously influence the temporal variations of the AOD in each province, since the differences in the emissions between provinces occur also due to the differences in regional policies on emission control and their implementation (Jin et al., 2016; van der A et al., 2017). In addition, both meteorological conditions and large-scale circulation will vary from year

to year and between different parts of China during each year. As a result, the aerosol properties and their spatial and temporal variations are expected to be different across China. As an illustration, Fig. 1 shows the AOD time series retrieved using the ATSR Dual View aerosol retrieval algorithm (ADV) version 2.31 (Kolmonen et al., 2016; Sogacheva et al., 2017), for the years 1995-2011, for two areas. One area covers mainland China, the other one only south-eastern (SE) China (see Sect. 2 and Fig. 2 for specification of these regions). Clearly, the AOD over SE China is substantially higher than over mainland China, but also the AOD increases much faster over SE China. Apart from that, the interannual variations are quite similar, with minima and

maxima occurring in the same years but with larger amplitudes over SE China.



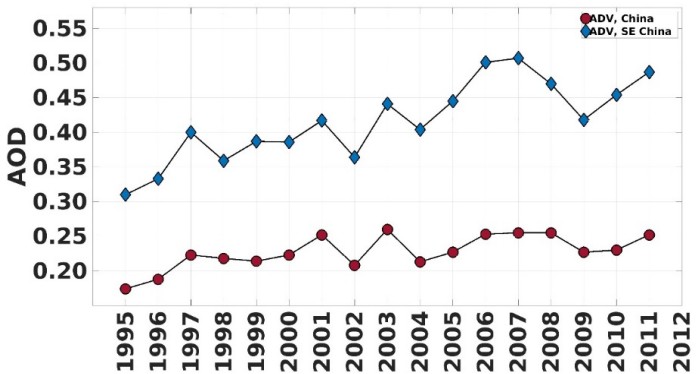

**Figure 1. Time series of ATSR-retrieved AOD at 550 nm over China for the years 1995-2011. Note that data are missing in the beginning of the ATSR-2 observation period in 1995 and 1996, and AATSR data start from August 2002.**

In this paper, the work presented in de Leeuw et al. (2018) is extended to obtain information on the occurrence of aerosols and their spatial and temporal variation over China with a focus on regional differences in annual and seasonal AOD behaviour for selected regions. In addition, the study period is extended by including 2016 and 2017 and the most recent update of the MODIS AOD data set, Collection 6.1 (C6.1), is used instead of C6. The C6.1 AOD validation results, the C6.1 vs C6 comparison, differences between ADV and MODIS C6.1 annual and seasonal AOD aggregates, as well as in

10 AOD tendencies during the overlapping period (2000-2011) are discussed. The results from the ADV and MODIS AOD comparison will be utilized in Part II to construct a combined long-term AOD time series from ADV and MODIS for the period of 1995-2017. AOD tendencies over the selected regions will be estimated in Part II for the different periods characterized by changes in air pollution control policies in China (Jin et al., 2016; van der A et al., 2017).

 The structure of this paper is as follows. In Sect. 2, the study area, including the selection of the 10 regions, is described. In Sect.3, satellite data and

15 AOD validation results are discussed, with a focus on the MODIS C6.1 and C6 AOD differences over China. In Sect. 4, the interannual variation is discussed based on AOD anomaly maps for the period 1995-2017 and annual AOD time series from ATSR (1995-2011) and MODIS (2000-2017); AOD tendencies in the overlapping period (2000-2011) are estimated and discussed. Sect. 5 focuses on the seasonal AOD variations and their differences for the selected regions across China. In Sect.6, the main results are summarised as conclusions.



## 2. Study area and selection of different regions

The study area encompasses the same area as in de Leeuw et al. (2018), i.e. between 18º-54º N and 73º-135º E, with a focus on mainland China, i.e. the area within the Chinese borders indicated by the blue line shown in Fig. 2. The spatial variations of the AOD (Fig.2) combined with geographical knowledge (cf. de Leeuw et al., 2018) and general knowledge of the locations of highly populated and industrialized urban centers in China was used to

select regions with different characteristics for a more detailed study on the long-term variation of the seasonal and annual AOD.  The results are expected to show differences in AOD across China due to different climate and differences in economic development. Such considerations resulted in the selection of 10 study areas as shown in Fig. 2: seven of them (regions 1-7) in SE China, one covering the Tibetan Plateau (region 8), one over the Taklamakan desert (region 9) and one over northeastern (NE) China (region 10). SE China defined in this study as the over-land area between 20º-41º N and 103º-135º E, includes areas 1-7. It is noted that all areas used in this study only consider the AOD over mainland China, i.e. AOD over the oceans or islands

is not included.

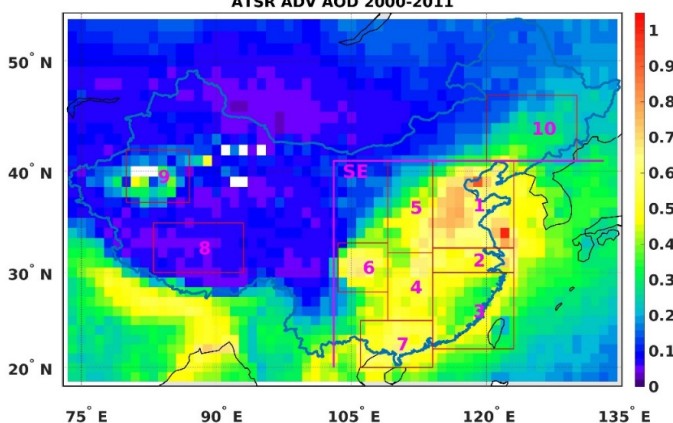

**Figure2. Regions over mainland China selected for further study of seasonal, interannual and long term behaviour of the AOD, overlaid on the ATSR-retrieved (ADV version 2.31) 12-year aggregated AOD map. Mainland China is indicated with the blue line. The figure shows 10 selected regions over China and a larger area over SE China indicated with SE.**

Obviously, other choices are possible, such as those made by Luo et al. (2014) or Wang et al. (2017). The regions selected by Luo et al. (2014) are overall somewhat smaller than those in Fig. 2 and some of them are shifted with respect to the choices made for this study. However, overall the choices are similar and seem to cover major urban/industrial regions such as BTH, YRD and PRD, Sichuan/Chongqing as well as cleaner regions in the north (region 10 in Fig. 2) and south-east (region 3). Also regions were chosen to represent the Tibetan Plateau and Taklamakan desert. Wang et al. (2017) selected




seven regions in North China (north of $32^o$ N) some of which partly overlap with the regions selected for the current study. Other studies on the seasonal variation over China were guided by the location of observational sites (e.g. Wang, Y., et al., 2011; Che et al., 2005; Wang, et al., 2015). Another choice could be by province (e.g., van der A et al., 2017). However, some provinces would cover a mix of high and low AOD regions, while other provinces would be too small for a statistically meaningful data set.

## 3. Satellite data

### 3.1 MODIS C6.1 DTDB and ATSR ADV version 2.31.

The data used in this work were discussed in detail in de Leeuw et al. (2018). However, in the current study, the MODIS C6 DTDB merged AOD was replaced with the recently released MODIS C6.1. In addition, MODIS/Terra data for 2016 and 2017 have been included in the analysis to provide the information on the AOD evolution in the most recent years. In short, L3 (averaged on a grid of $1^o$x$1^o$) AOD data retrieved from ATSR-2 (1995-2002) and AATSR (2002-2012) (together referred to as ATSR) using ADV version 2.31 (Kolmonen et al., 2016; Sogacheva et al., 2017) and MODIS/Terra AOD C6.1 merged DTDB (L3) data (2000-2017, https://ladsweb.modaps.eosdis.nasa.gov/ ) were used together to cover the period from 1995-2017. Hereunder, the ATSR ADV version 2.31 AOD product will be referred to as ADV. The MODIS/Terra AOD C6.1 merged DTDB AOD product will be referred to as MODIS.

In this study, the annually averaged AOD data were obtained by averaging monthly aggregated AOD data in each year. Furthermore, the seasonal means were obtained as averages of monthly aggregates for winter (DJF, including December, January, and February), spring (MAM, including March, April, and May), summer (JJA, including June, July, and August) and autumn (SON, including September, October, and November). Annual and seasonal linear AOD tendencies for both MODIS and ADV AOD for the overlapping period (2000-2011), when both ATSR and MODIS exist, were estimated using the least-squares linear regression method (Chandler and Scott, 2011).

### 3.2 Comparison between MODIS merged DTDB C6.1 and C6 AOD

In MODIS C6.1, the brightness temperatures biases and trending were significantly reduced compared to C6 affecting ice cloud detection over water surfaces (https://modis-atmos.gsfc.nasa.gov/sites/default/files/ModAtmo/C6.1_Calibration_and_Cloud_ Product_Changes_UW_frey_CCM.pdf, last accessed 28.02.2018). The electronic crosstalk correction discussed in Wilson et al. (2017) was made which greatly improves the performance of the cloud mask.

The difference between C6.1 and C6 annual AOD over China averaged for the period 2000-2011 is shown in Fig. 3. This period was chosen because of the overlap between ATSR and MODIS, which will be studied in the current manuscript. Over most of China the difference between C6.1 and C6 is very



small, as shown in Fig. 3, except for certain areas. The annual aggregated C6.1 AOD over the Tibetan Plateau and over the area north from the Taklamakan desert is 0.1-0.2 lower than for C6, while over Ningxia province (ca. 35-37°N and 103-107°E) and the Sichuan basin (ca. 28-30°N and 103-107°E) the AOD has increased by 0.1-0.2. Fig. 3b shows that the AOD differences over the TP and in the north are mostly due to the lower C6.1 AOD in the winter (DJF, about 0.15) and spring (MAM, up to 0.25) and over Ningxia due to the much higher AOD in these seasons. Over the Sichuan basin the C6.1 and

5  C6 AOD is similar in all seasons except in winter when C6.1 is about 0.25 higher. Similar changes in the AOD between C6.1 and C6 are shown by A. Sayer (https://www.atmos-chem-phys-discuss.net/acp-2017-838/acp-2017-838-RC1-supplement.pdf) for the period 2000-2008 over China.

As regards coverage, over most of mainland China, the differences between C6.1 and C6 are very small, except over the elevated areas such as the Tibetan Plateau, the Huangiu Gaoyuan Plateau and areas in the NW and NE of China (A. Sayer, personal communication, 2017). However, the MODIS AOD coverage has increased over other locations, which is concluded from the increasing number of points available for validation, as discussed below in

10  Sect. 3.3.1.



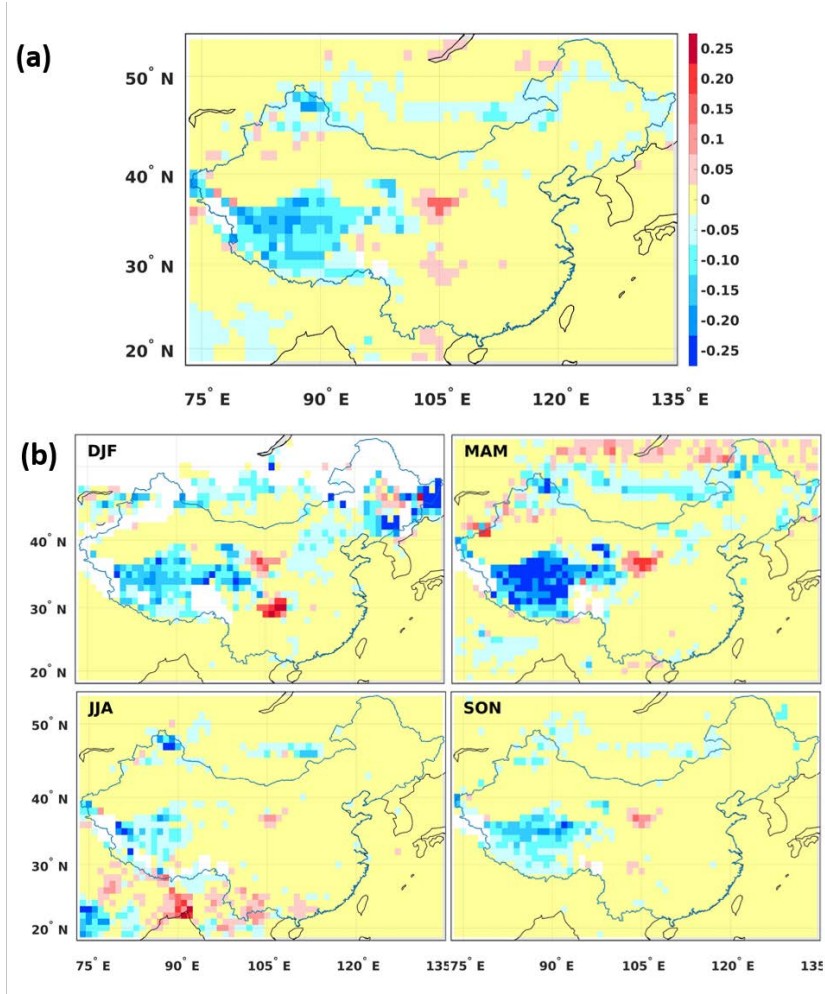

**Figure 3. Difference between MODIS C6.1 and C6 DTDB merged AOD over China: annual averages for 2000-2011 (a) and seasonal averages for the same period (b, where DJF – winter, MAM – spring, JJA – summer and SON - autumn). Areas for which no data are available are shown in white.**



### 3.3 ADV and MODIS AOD validation

### 3.3.1 General validation over China and validation results for selected regions.

MODIS C6.1 validation with AERONET sites in the study area (Fig. 4, right) has been performed similar to C6 validation, as described in de Leeuw et al. (2018). Briefly, collocated data are used, i.e. satellite data within a circle with a radius of 0.125° around the AERONET site are averaged and compared

5  with the averaged AERONET data measured within ±1 hour of the satellite overpass time (Virtanen et al., 2018). The amount of validation points in C6.1 is higher than that in C6 (4962 against 4760, respectively). The correlation coefficient (R) has increased from 0.90 to 0.92 in C6.1. Bias and root-mean-square error (rms) have decreased from 0.007 to 0.06 and from 0.23 to 0.2, respectively.  Thus, in a statistical sense MODIS C6.1 performs slightly better than C6.

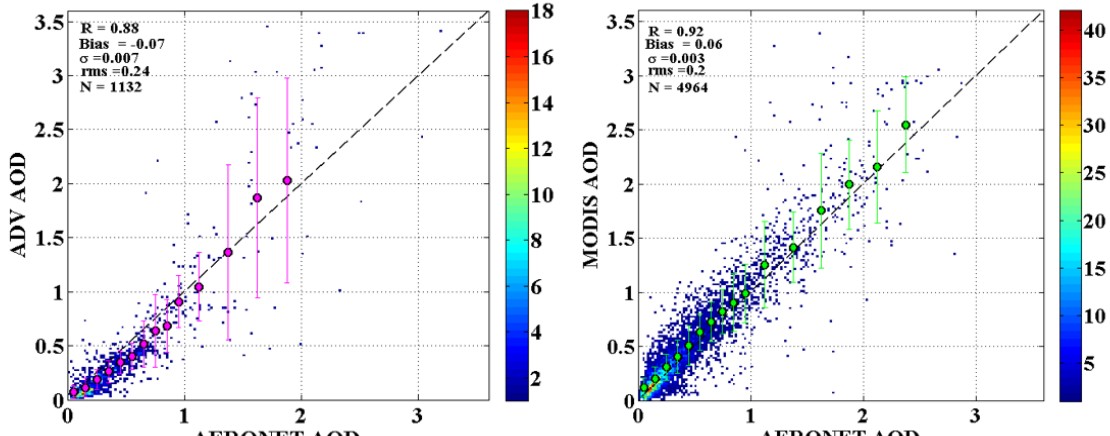

10  **Figure 4. Density scatterplot of ATSR ADV v2.31 AOD (left, reproduced from de Leeuw et al., 2018, Figure 7), and MODIS C6.1 DTDB (right) versus AOD from AERONET stations in mainland China for the years 2002-2011. The filled circles are the averaged ATSR AOD binned in 0.1 AERONET AOD intervals (0.25 for AERONET AOD>1.0) and the vertical lines on each circle represent the 1-sigma standard deviation of the averages. Statistics in the upper left corner indicate correlation coefficient R, bias, standard deviation, root-mean-square (rms) error and number of data points (N). The colour bar on the right indicates the number of data points.**

However, the main difference in validation results for ADV and MODIS, the bias, which is similar in absolute value but opposite in sign (0.06 for MODIS and -0.07 for ADV), still exists (Fig.4). This difference in AOD bias is emphasised here because it explains the difference in AOD between ADV and MODIS, as shown and discussed below, and will be used in Part II to construct the ADV and MODIS combined time series.



We also checked whether AOD validation results differ across China, where aerosol conditions are changing considerably from region to region, reflecting differences in primary and secondary aerosol sources, population density, industry, etc. Unfortunately, AERONET stations are sparsely located in China and long-term measurements have been conducted for few locations only (see Table 1 and Fig. 1 in de Leeuw et al., 2018), mostly in SE China.

The validation statistics for the selected regions, where AERONET AOD data are available, are shown in Table 1 (note the low number of validation

5  points N in regions 5, 7, 8 and 10). For both ADV and MODIS, R was rather high (0.84-0.92) for all regions presented in the analysis, except for region 8, where correlation with AERONET was much lower for both data sets (0.33 and 0.35, ADV and MODIS respectively). In region 8, which includes the sparsely populated Tibetan Plateau which is often covered with snow, AOD is very low and high uncertainties in AOD are expected related to the retrieval algorithms limitations (e.g., Kolmonen et al., 2016, Sayer et al., 2014). In region 5, both ADV and MODIS show a strongly negative AOD bias (-0.30 and -0.15, respectively). A high positive AOD bias (0.16) is obtained for MODIS in region 7.

**Table 1. AOD validation results (number of points (N), correlation coefficient (R), bias, standard deviation ($\sigma$) and root-mean-square (rms) error) for ADV and MODIS (MOD) obtained for the regions (left column), where AERONET data are available.**

| region | N ADV | N MOD | R ADV | R MOD | bias ADV | bias MOD | σ ADV | σ MOD | rms ADV | rms MOD |
|---|---|---|---|---|---|---|---|---|---|---|
| China | 1132 | 4964 | 0,88 | 0,92 | -0,07 | 0,06 | 0,07 | 0,003 | 0,24 | 0,20 |
| China, SE | 1074 | 4846 | 0,88 | 0,92 | -0,07 | 0,06 | 0,007 | 0,003 | 0,25 | 0,20 |
| 1 | 475 | 2928 | 0,89 | 0,94 | -0,08 | 0,06 | 0,014 | 0,003 | 0,30 | 0,20 |
| 2 | 118 | 188 | 0,86 | 0,84 | -0,09 | 0,00 | 0,023 | 0,024 | 0,26 | 0,35 |
| 3 | 343 | 937 | 0,84 | 0,89 | 0,00 | 0,07 | 0,009 | 0,005 | 0,16 | 0,15 |
| 5 | 15 | 80 | 0,90 | 0,87 | -0,30 | -0,15 | 0,049 | 0,014 | 0,22 | 0,19 |
| 7 | 9 | 18 | 0,92 | 0,92 | -0,01 | 0,16 | 0,006 | 0,032 | 0,17 | 0,24 |
| 8 | 21 | 11 | 0,37 | 0,33 | 0,04 | 0,02 | 0,011 | 0,017 | 0,05 | 0,06 |
| 10 | 11 | 26 | 0,88 | 0,96 | 0,05 | 0,04 | 0,073 | 0,019 | 0,32 | 0,10 |

### 3.3.2 ADV and MODIS collocated points annual and seasonal validation

15  The validation results presented in Fig. 4 show that MODIS AOD is positively biased and, in contrast, ADV is negatively biased. However, since more validation points exist for MODIS than for ADV (Table 1), which is likely explained by better MODIS coverage (see the discussion on the ADV and MODIS coverage below in Sect.3.4), a direct comparison of the algorithms performance to show differences in the retrieved AOD cannot be made. Instead, the retrieval performance was evaluated using collocated ATSR-MODIS/Terra-AERONET data. Thus, only those AOD data are used for



validation, when both MODIS and ADV have achieved a successful retrieval over AERONET sites and the overpasses were within ±90 min while also AERONET data were available. In total, 255 collocated points have been recognized for the period 2002-2011.

To compare the ADV and MODIS performance for the cases when both ADV and MODIS provide a successful retrieval, we carry out the ADV and MODIS AOD validation for the collocated points for the whole period (Fig. 5, upper panel) and also for each of the seasons (Fig. 5, middle panel for ADV and lower panel for MODIS). The scatter plots of the ADV or MODIS AOD versus AERONET AOD show that for all collocated points (Fig. 5, upper panel) R is similar for ADV and MODIS (0.92 and 0.93, respectively), while bias is negative for ADV (-0.11) and positive for MODIS (0.06). In winter, when the number of collocated points is low (10), both ADV and MODIS slightly underestimate AOD. In spring, R is lower than for other seasons and is the same (0.81) for both ADV and MODIS, while bias is 0 for ADV and positive for MODIS (0.11). In summer, R is slightly higher for MODIS (0.96 against 0.94 for ADV). Bias is similar in absolute value (0.13) but has different sign for ADV (positive) and MODIS (negative) in summer. In autumn, ADV performs slightly better (R is 0.92 and 0.88, bias is -0.02 and 0.05 for ADV and MODIS, respectively). Thus, is all seasons except winter, positive bias is observed for MODIS, while ADV AOD has negative compared to AERONET bias in all seasons except spring, when ADV bias is 0.



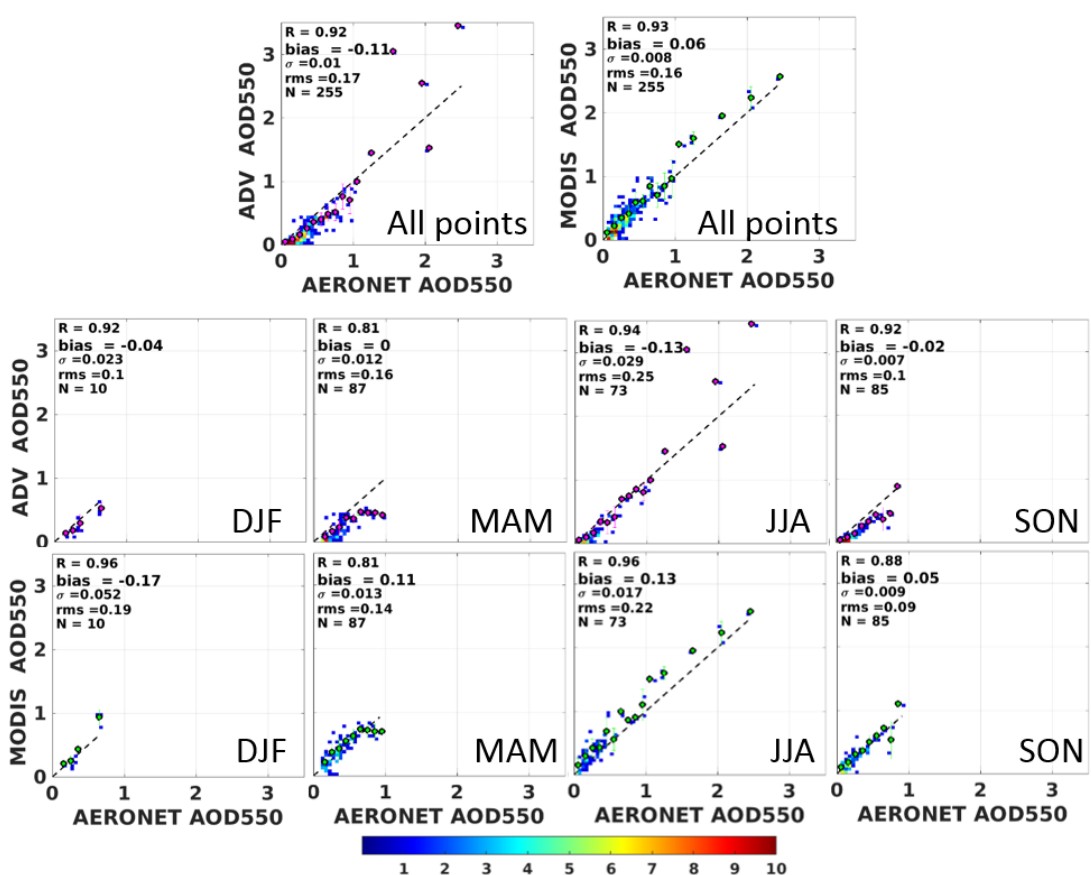

**Figure 5. Density scatterplots of collocated ATSR ADV v2.31 AOD and MODIS C6.1 DTDB merged AOD versus AOD from AERONET stations in China for the years 2002-2011: all points (upper panel) and seasonal statistics (middle panel for ADV and lower panel for MODIS). The colour bar at the bottom indicates the number of data points.**





We also checked if the difference in the ADV and MODIS AOD depends on the difference in overpass time between ATSR and MODIS/Terra. The validation results for the ADV/MODIS/AERONET collocated points are shown in the scatterplot of MODIS AOD versus ADV AOD in Fig. 6. The color code indicates the difference in the exact overpass time. For all collocated points, MODIS AOD is usually higher, with a bias of 0.2. That positive difference does not depend on the difference in overpass times between ATSR and MODIS/Terra and thus cannot be explained by the influence of the

possible AOD daily cycle.

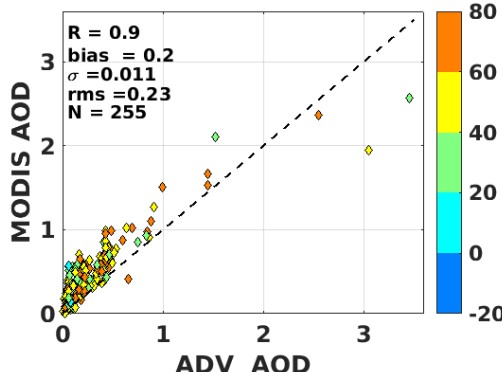

**Figure 6. MODIS/Terra C6.1 DTDB merged AOD versus ATSR ADV v2.31 AOD, for collocated ATSR-MODIS/Terra-AERONET data, as described in the text of Sect. 3.3.2. The colours (scale at the right) indicate the difference between the MODIS/Terra and ATSR overpass times in minutes.**

**3.4 ADV and MODIS coverage over selected regions**

As introduced in de Leeuw et al. (2018), ATSR and MODIS have different temporal and spatial coverage over China. In brief, MODIS/Terra covers China in 1-2 days, while with ATSR China is covered in 4-5 days.

ADV data sets for the years 1995, 1996 and 2012 are incomplete. For 1995 and 1996, ADV AOD data are available for the second half of each year (June-December and July-December, respectively). However, all available ADV AOD data in 1995 and 1996 are used in current study to construct the annual aggregates for comparison with other annual aggregates. Obviously, the 1995 and1996 aggregates are not exact and therefore the possible error

related to the missing data has been estimated by comparison of the full-year (January to December) AOD composites with the half-year (July-December) AOD composites for the complete years (1997-2011). This comparison shows that when the half-year aggregate was used to present the full-year aggregate, the AOD was underestimated by, on average, –0.036 (with standard deviation of 0.02), or about 15% of the yearly aggregated AOD value. In this study, the aggregated AOD for the years 1995 and 1996 have not been corrected for the missing data and those years are included in the further analysis as they are. Another point worth mentioning is the white area in the far west of the study area (Fig.8) where ATSR-2 did not provide data because



the data collection was switched off for data transfer to the receiving station over that area. For 2012, the ADV AOD data are available until the connection with the satellite was lost on the 6th of April

For MODIS/Terra, the AOD data record starts from end of February 2000. Thus, the winter season for 2000 is missing.

To estimate the spatial coverage of AOD, the fraction of the area where AOD is available has been calculated for all seasonal and annual aggregates for
the selected regions (Table S1, Supplement). In spring (MAM), summer (JJA) and autumn (SON), the ADV coverage reaches 84%, 91% and 91%, while MODIS coverage is 93%, 93% and 97%, respectively, over mainland China. Throughout the year, both ADV and MODIS coverages are close to 100%, except for region 9, where the ADV coverage is 62%. For both ADV and MODIS, the Tibetan Plateau (region 8), the Taklamakan desert (region 9) and NE China (region 10) are covered less compared with other regions, all over the year.

 Over the seasons, the lowest coverage is observed in winter (DJF), when northern and western China are covered with snow. As most aerosol retrieval
algorithms, MODIS and ADV have difficulty retrieving AOD over snow and ice (Hsu et al., 2013; Istomina et al., 2011; Kolmonen et al. 2016), as well as all year round over bright surfaces such as the Taklamakan desert. On average, in winter MODIS provides AOD over 70% of mainland China, while ADV AOD is available over 35% of China. For certain years, ADV AOD is not available in the winter over regions 9 and 10 and thus not shown in the analysis (Sect. 5). However, since the annual AOD time series for ADV and MODIS in regions 9 and 10 show similar tendencies (discussed later in Sect. 5.2.3), we assume that missing (for some years) ADV AOD in winter does not bias the results considerably.

Thus, besides the difference in the validation results presented and discussed above (Sect.3.3), which is likely due to the differences in the ADV and MODIS AOD retrieval approaches and their implementation, the difference in the ATSR and MODIS/Terra spatial and temporal coverage might influence the AOD composites. Another exercise might be performed, where AOD aggregates are built for collocated ADV and MODIS pixels, but this is out of the scope of the current paper, where AOD aggregates for all points available in the ADV and MODIS standard products are analysed and compared.

## 4. AOD interannual variations

For the discussion of the spatial distribution of the AOD over China and its temporal evolution we refer to SE China as defined in Sect. 2, which roughly encompasses the high AOD area east of the Tibetan Plateau (cf. Fig. 2), and SE from the line connecting the TP with the west of Hebei. The area to the west of that line, roughly the area where the AOD is overall smaller than over SE China, with exception of the deserts during dust events, is referred to as west China.

Figure 7 shows the spatial distribution of the ADV and MODIS C6.1 AOD aggregated over the years 2000-2011 across China. For that period, both
ATSR and MODIS/Terra AOD data are available, which allows for a proper inter-comparison of the AOD for this period. In brief, high AOD with values between 0.5 and 0.9 is observed with ADV over SE China, which is the most densely populated and industrialized area. The lowest values are observed in sparsely populated areas with low industrial aerosol emissions over Tibet and Inner Mongolia, where AOD is smaller than 0.1. AODs of about 0.3 are



observed over NE China (Fig. 7, left). Over west China, the AOD is high over the Taklamakan desert, but over the bright desert surface the ADV does not produce reliable data (Kolmonen et al., 2016) and thus for such pixels AOD is underestimated or no ATSR values are presented (the white areas in Fig. 7, left). MODIS L3 data (Fig. 7, middle) shows that the 12-year aggregated AOD over this area is in the range 0.4-0.6. It is further noted that the expected high AOD due to dust emitted over the Gobi desert (GD) is not detected in neither ADV nor the MODIS data. The ADV and MODIS AOD

5 annual aggregates for different areas will be discussed in detail in Section 5.

The ADV-MODIS difference map in Fig. 7 (right), shows that MODIS AOD is generally higher than that from ADV. This was extensively discussed in de Leeuw et al. (2018), based on comparison of ADV and MODIS C6 DTDB merged AOD data with AERONET. Figure 7 (right) shows that for C6.1 the differences between ADV and MODIS are similar over most of China. However, over the Tibetan Plateau, where MODIS C6.1 AOD has decreased compared to C6 (Fig.3), the difference between ADV and MODIS has also decreased.

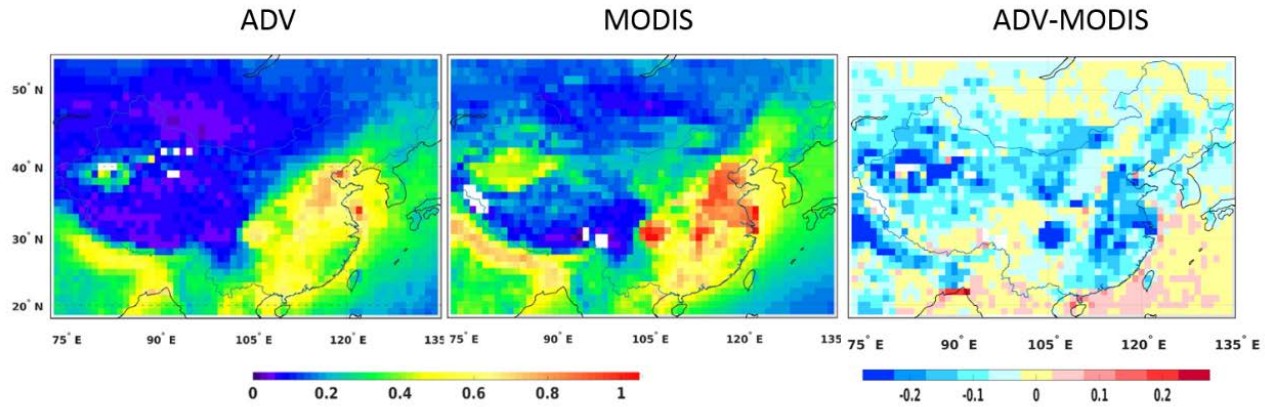

**Figure 7. Multi-year aggregated AOD for the period 2000-2011 from ATSR ADV v2.31 (left column), MODIS/Terra C6.1 merged DBDT (middle) and the difference ATSR-MODIS (right). (ADV AOD map is adapted from de Leeuw et al. (2018), Fig. 2).**

### 4.1 Evolution of AOD over China: annual anomaly maps

To study the temporal evolution of the AOD over China, the anomaly maps have been created by subtracting the multi-year aggregates (2000-2011, i.e.

15 the period when ATSR and MODIS/Terra both provided complete years of data) from the yearly aggregates. The evolution of the AOD can be followed for different regions, and regions with different evolution of AOD can thus be identified. These series of maps are complementary to time series showing the variation of the AOD averaged over selected regions, as will be presented and discussed Section 4.2 and in Part II.

Three sets of anomaly maps have been created:



- For the pre-EOS (Earth Observing System) period (1995-2000) when only ATSR-2 was available. As discussed above, the 1995 and 1996 aggregates might be underestimated from the true value by about –0.036 (with the standard deviation of 0.02)
- For the ATSR-MODIS/Terra overlaping period 2000-2011: MODIS and ADV yearly anomaly maps (with respect to the 2000-2011 average for each instrument)
- For the post-Envisat period, when only MODIS/Terra data was available, i.e. 2012-2017.

The anomaly maps for ADV and MODIS for the period 1995-2017 are presented in Fig. 8. For the overlapping period, both instruments show similar patterns of AOD tendencies (see also Fig. 9 and discussion in Sect. 4.2). Thus, the anomaly maps for MODIS AOD are shown for the overlapping period (Fig.8).

Strong negative anomalies over SE China and in particular over the Himalayas and the north of India and Bangladesh during 1995-2001 show that during
that period AOD was lower compared to AOD averaged over 2000-2011. Towards 2006, the AOD anomalies were becoming less negative: yearly AOD was increasing and difference between yearly AOD and AOD averaged over 2000-2011 was decreasing. Starting from 2006, a positive AOD anomaly, which show that yearly AOD was higher than AOD averaged over 2000-2011, is observed over SE China. Between 2006 and 2011, the positive anomalies were increasing, showing year-to-year increase of AOD. Starting from 2011, the positive anomaly is decreasing and changing sign to negative, which show that AOD is starting to decrease. Starting from 2015 the anomaly is overall negative and increasing gradually towards 2017, which is the last year
in our studies.

However, there is no homogenous AOD increase and the patterns vary over different areas of southeast China. Starting from the BTH area (region1), the anomalies are strongly negative until 2001 after which the anomalies disappear (with 2004 as an exception) and turn to positive with a maximum in 2006. Thereafter the AOD anomaly remains positive and at about the same level. In contrast to the BTH area, in the southeast of the high AOD study area (region 3) the AOD increase sets in much earlier with a positive anomaly in 1997 and neutral values until 2000. From 2000, a clear increase is
observed in the following years until 2009 when the AOD anomaly drops to around zero or even negative in 2010. After 2010, positive and negative years alternate. From 2006, the AOD anomaly in the south is consistently smaller than in the BTH area. In the intermediate area over the YRD (region 2) encompassing large urban and industrial developments such as Shanghai and Nanjing, the multi-year average appears to be quite representative for the AOD with negative anomalies until 2000, after which the anomalies fluctuate around zero with neither very high nor very low values.

Similar developments are observed over the Sichuan/Chongqing area (region 6) with a strong positive anomaly in 1997 and 1999, followed by negative
anomalies in 2000-2002. After 2002, no significant deviations are observed until 2006 when the AOD exceeds the multi-year average somewhat and this situation remains, with small fluctuations until 2014 when a strong negative anomaly is observed which is stronger and spreads over the wider region of SE China during the next (2015-2017) years.



A "belt" of positive AOD anomaly is observed east from the Taklamakan desert in 2001, which is likely the year of the most intensive dust transport during the period of interest. The widespread positive anomaly in 1998 and 2003 in the NE of China is likely the consequence of forest fires over Russia. This is most notable in region 10, but also regions 1 and 5 show the enhanced AOD in 2003.

The AOD tendencies across China will be discussed in more detail in Part II in relation to the emission regulation policies in China.





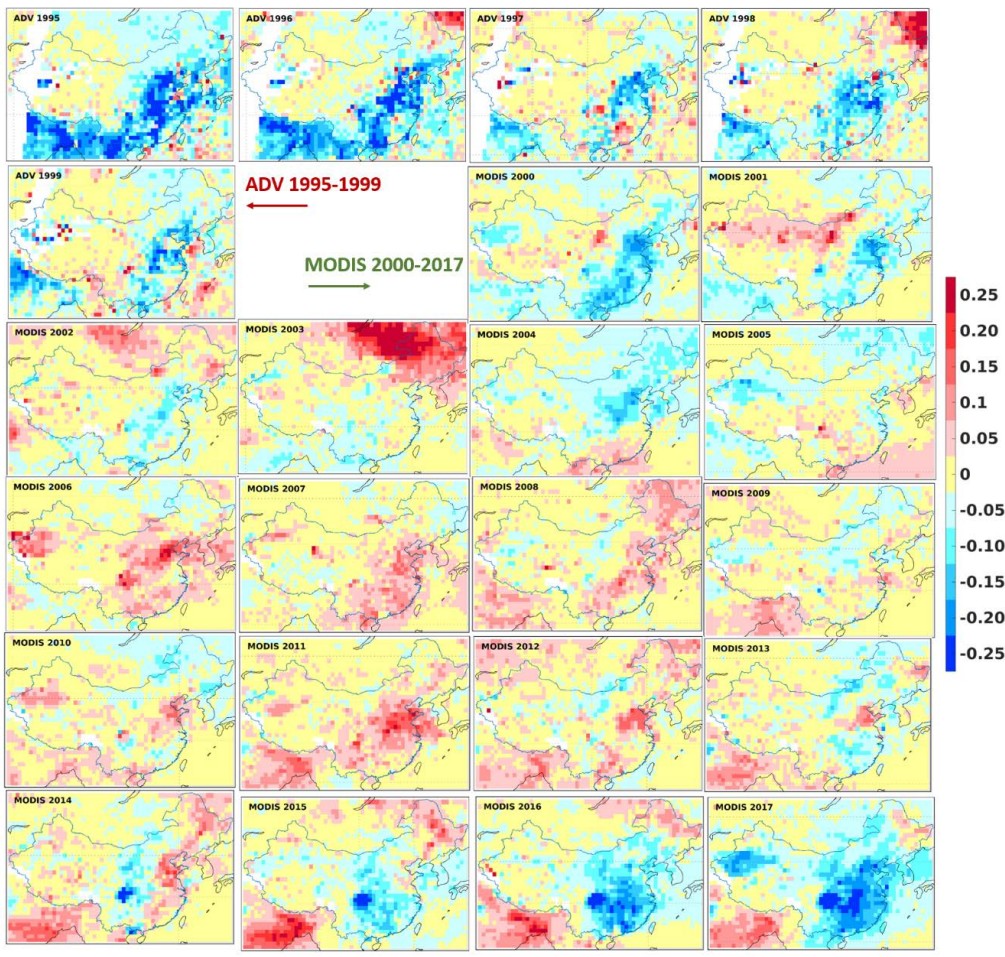

**Figure 8. AOD anomaly maps for the years 1995-2017 calculated for ATSR ADV (1995-1999) and MODIS (2000-2017). Anomalies were calculated for each year by subtracting the multiyear (2000-2011) average for correspondent instrument from the annual aggregate.**





**4.2. Long-term annual time series: ATSR (1995-2011) and MODIS (2000-2017)**

Figure 7 shows the spatial distribution of the multi-year averaged ADV and MODIS AOD for the years 2000-2011. However, this leaves out the pre-EOS period covered by ATSR-2 and the post-Envisat period covered by MODIS. Furthermore, inter-annual variations occur, as shown in Fig. 8. These variations will be discussed based on the multi-year annually averaged AOD time series in Fig. 9, which include the periods 1995-2011 for ADV and

2000-2017 for MODIS, for China, SE China and for each of the 10 regions. In this figure, the AOD datasets are divided into three periods, i.e. pre-EOS with only ATSR-2 (1995-2000), post-Envisat with only MODIS/Terra (2011-2017), and the overlap period (2000-2011, shaded light green) when both algorithms provided valid AOD retrievals. We choose the overlapping period to see whether both instruments show similar AOD tendencies or whether the AOD tendencies for ADV and MODIS are different. AOD tendencies for the whole 1995-2017 period are discussed in detail in Part II. Briefly, the time series of the yearly averaged AOD over China (Fig. 9) show a small increase of the AOD over the years 1995-2011, with a somewhat larger tendency

for MODIS than for ATSR, whereas from 2011 the MODIS data show a definite decrease. This behavior seems to be mainly determined by that in SE China (and regions 1-7 therein) where the AOD is substantially higher, and tendencies until 2011 have a similar direction but are much stronger, than over the west and north of China (regions 8-10). After 2011, the MODIS AOD shows a distinct decrease.

For the overlapping period, linear fits were made using a MATLAB tool (https://se.mathworks.com/products/matlab.html and detailed description of the statistics) to determine the variation of the AOD versus time. AOD tendencies (dAOD) per decade, bias and slope for the linear regression lines, as well

as p-value estimated with the t-test and absolute error for linear fits are presented in Table 2 (for annual aggregates) and Table S2 (for seasonal aggregates) for all selected regions.

For both China and SE China, the annual (Fig. 9) and seasonal (Figs. S1-S4) time series for ADV and MODIS are very similar, albeit with an almost constant offset with MODIS high and ADV low. When looking at the long-term time series of the yearly averaged AOD for each of the 10 regions, this behavior is replicated, with some anomalous years for each of them. The possible exception is region 8 (the Tibetan Plateau) where the AOD is very low

in comparison with other regions with practically no interannual variation or long-term tendency.





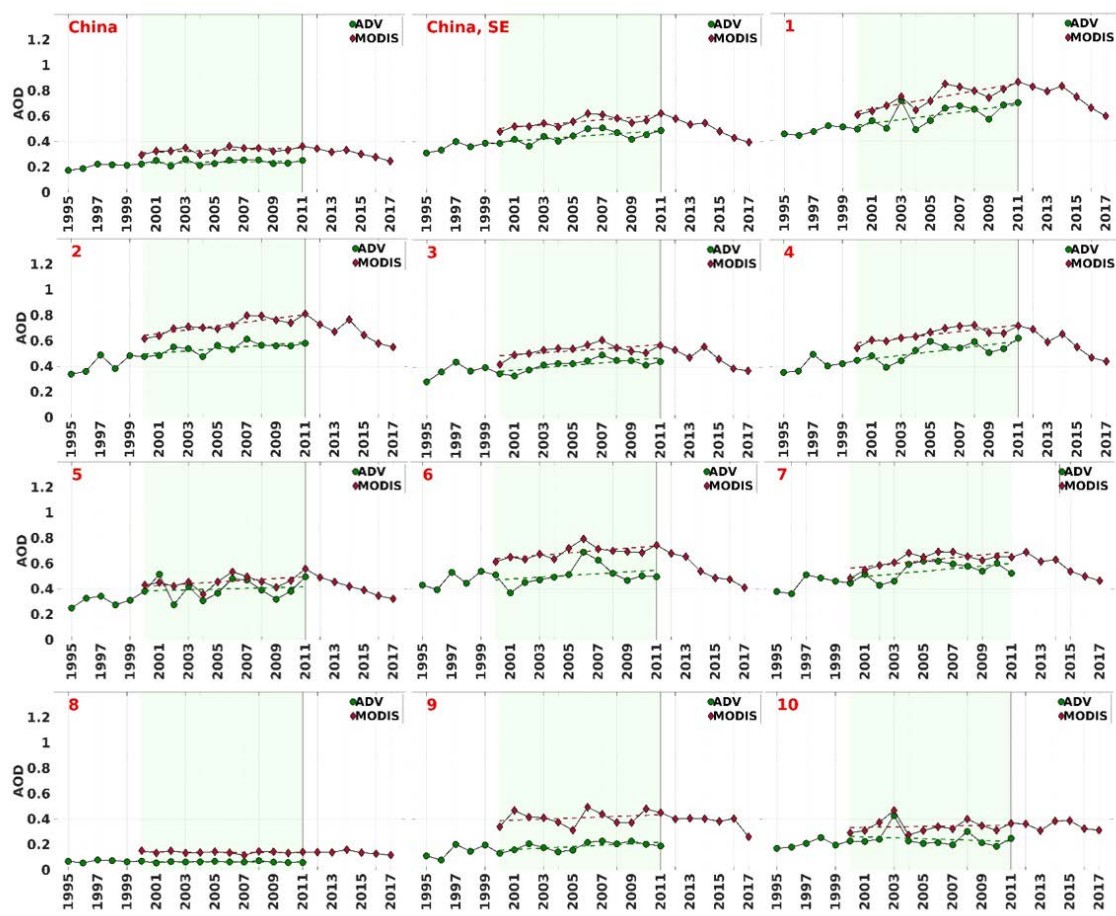

**Figure 9. Time series of the yearly averaged AOD over China, SE China and each of the 10 selected regions, for ADV (1995-2011, green circles) and MODIS (2000-2017, red circles). The overlapping period is colored with light green. AOD linear fits for the overlapping periods are shown for each instrument (green and red dashed lines for ADV and MODIS, respectively). Statistics for linear fits are shown in Table 2.**



For the overlapping period, only region 8 shows a slightly negative yearly AOD tendency and region 10 shows opposite in sign AOD tendencies for ADV (negative) and MODIS (positive). The strongest yearly AOD positive tendency has been obtained with both instruments in regions 1-4 in SE China (0.148/0.201, 0.083/0.149, 0.095/0.078, 0.139/0.127 for ADV/MODIS and regions 1, 2, 3 and 4 respectively). All those positive tendencies were estimated with the p-values < 0.05 and an absolute error smaller than 7%. For SE China, similar AOD tendencies were estimated with ADV and MODIS
(0.082 and 0.096, respectively). For other regions, the AOD tendencies are slightly positive with somewhat higher numbers obtained for MODIS. For the whole country, indicated as China, both instruments show similar AOD tendencies, with an AOD increase in the overlapping period of 0.013 for ADV and 0.035 for MODIS. This difference can be explained by the ADV and MODIS validation results, where overall the MODIS-retrieved AOD is somewhat higher than that retrieved with ADV.

**Table 2. AOD tendency (dAOD) per decade and statistics for the annually combined AOD time series linear fit (p-value, bias, slope and absolute error (ae, %)) for the overlapping period (2000-2011) for different regions (r, where region 11 is the whole mainland China and region 12 is mainland SE China). dAOD is highlighted in bold if the correspondent p-value <0.05.**

| region | ADV | | | | | MODIS | | | | |
|---|---|---|---|---|---|---|---|---|---|---|
| | dAOD /decade | p-value | bias | slope | ae, % | dAOD /decade | p-value | bias | slope | ae, % |
| China | 0,013 | 0,4132 | -2,43 | 0,001 | 6,1 | 0,035 | 0,0682 | -6,75 | 0,004 | 4,9 |
| China, SE | **0,082** | 0,0222 | -15,91 | 0,008 | 6,4 | **0,096** | 0,0046 | -18,78 | 0,010 | 4,5 |
| 1 | **0,148** | 0,0289 | -29,17 | 0,015 | 9,1 | **0,201** | 0,0010 | -39,64 | 0,020 | 5,6 |
| 2 | **0,083** | 0,0108 | -16,20 | 0,008 | 4,6 | **0,149** | 0,0002 | -29,13 | 0,015 | 3,4 |
| 3 | **0,095** | 0,0064 | -18,69 | 0,010 | 6,2 | **0,078** | 0,0431 | -15,17 | 0,008 | 5,9 |
| 4 | **0,139** | 0,0074 | -27,32 | 0,014 | 7,5 | **0,127** | 0,0010 | -24,82 | 0,013 | 4 |
| 5 | 0,029 | 0,6810 | -5,33 | 0,003 | 15,7 | 0,069 | 0,1275 | -13,45 | 0,007 | 8,5 |
| 6 | 0,070 | 0,3301 | -13,47 | 0,007 | 12,5 | **0,091** | 0,0258 | -17,61 | 0,009 | 4,7 |
| 7 | 0,104 | 0,0637 | -20,40 | 0,010 | 8,6 | **0,118** | 0,0153 | -23,00 | 0,012 | 6 |
| 8 | -0,002 | 0,5667 | 0,56 | 0,000 | 6 | -0,006 | 0,4978 | 1,32 | -0,001 | 5,6 |
| 9 | **0,058** | 0,0297 | -11,41 | 0,006 | 11,6 | **0,042** | 0,4006 | -8,10 | 0,004 | 11 |
| 10 | -0,035 | 0,5447 | 7,30 | -0,004 | 21,5 | 0,017 | 0,7160 | -3,15 | 0,002 | 12,6 |

**5. AOD seasonal variation**

The AOD over China varies not only in space but also seasonal variations are observed, as briefly discussed in de Leeuw et al. (2018) based on MODIS/Terra C6 data. Seasonal AOD maps for ADV and MODIS C6.1, aggregated over the years 2000-2011, and ADV-MODIS difference maps for each season, are presented in Fig. 10. The spatial distribution of seasonally averaged AOD is similar to the spatial patterns of the annually averaged AOD.



However, Fig. 10 shows some clear differences between ADV and MODIS, i.e. the MODIS AOD is often higher than that from ADV and MODIS has better coverage over bright surfaces. The latter is particularly prominent for the winter season (DJF) when the north and west of China are covered with snow. As mentioned before, like most radiometers used for aerosol retrieval, ADV has difficulty retrieving AOD over snow and ice, as well as all year round over bright surfaces such as desert areas. In other seasons than winter, ADV has reasonable coverage over most of China (see Table S1), except

the Taklamakan desert where high dust episodes are missed. It is noted that also MODIS does not provide AOD over snow and ice (Levy et al., 2013; Hsu et al., 2013), but over bright desert surfaces the DB algorithm does provide AOD (Hsu et al., 2003), which is included in the DBDT product used here. However, as shown in Fig. 10, MODIS also misses AOD over the Tibetan Plateau along the southern border of China during all seasons, as well as along the NW border in the winter. North of ca. 45ºN both MODIS and ADV do not provide AOD data in the winter.

As regards the AOD differences between MODIS and ADV, the difference maps in Fig. 10 show that MODIS is much higher (≥0.25) than ADV over

part of SE China in winter and spring, especially over the NCP and the Sichuan basin, as well as over the desert areas west of the Loess Mountains. In summer, these differences are overall much smaller (≤ 0.15-0.2) except over the Sichuan basin and Taklamakan and Gobi deserts and some smaller areas in SE China. Also just south of the Himalayas the MODIS AOD is much higher (≥0.25) than that retrieved using ADV. In autumn, the differences between MODIS and ADV are generally small (≤ 0.1) except for some regions in the SE of China (e.g. Sichuan, YRD and Hebei), as well as SW of the Himalayas. These observations on the differences between ADV and MODIS can be partly explained by the validation results presented in Fig. 4 and

Fig. 5, i.e. MODIS is biased high and ADV is biased low by a similar amount. However, these biases do not explain the seasonal variations of the differences between MODIS and ADV. Likely these are due to retrieval assumptions as regards the aerosol properties and the surface reflectance. The largest discrepancies are observed to the north of about 27º N and over relatively bright areas in the deserts as well as over the NCP, which is dryer in winter and spring than during the summer and autumn. Also, these regions are influenced by the desert dust with relatively large contributions to the AOD in spring as discussed below. ADV does not provide a quality retrieval over bright surfaces, but also for the SE China the ADV AOD is substantially

lower than that from MODIS. The substantially lower ADV-retrieved AOD in spring may indicate that the ADV retrieval of dust, which is most prominent in spring (e.g., Proestakis et al., 2018), needs some improvement. The difference between ADV and MODIS AOD may further be due to the fact that MODIS provides more results over bright surfaces, where ADV AOD is lower, if retrieved. In contrast, in summer the differences are much smaller.

As regards the AOD seasonal variation, the maps in Fig. 10 show similar variations for the ADV and MODIS-retrieved AODs. For instance, for the PRD the AOD is highest in spring and lower in other seasons whereas over the NCP in the area from BTH to the YRD the AOD is highest in summer. The

ADV and MODIS AOD seasonal aggregates for different areas will be discussed in detail in Section 5.1.



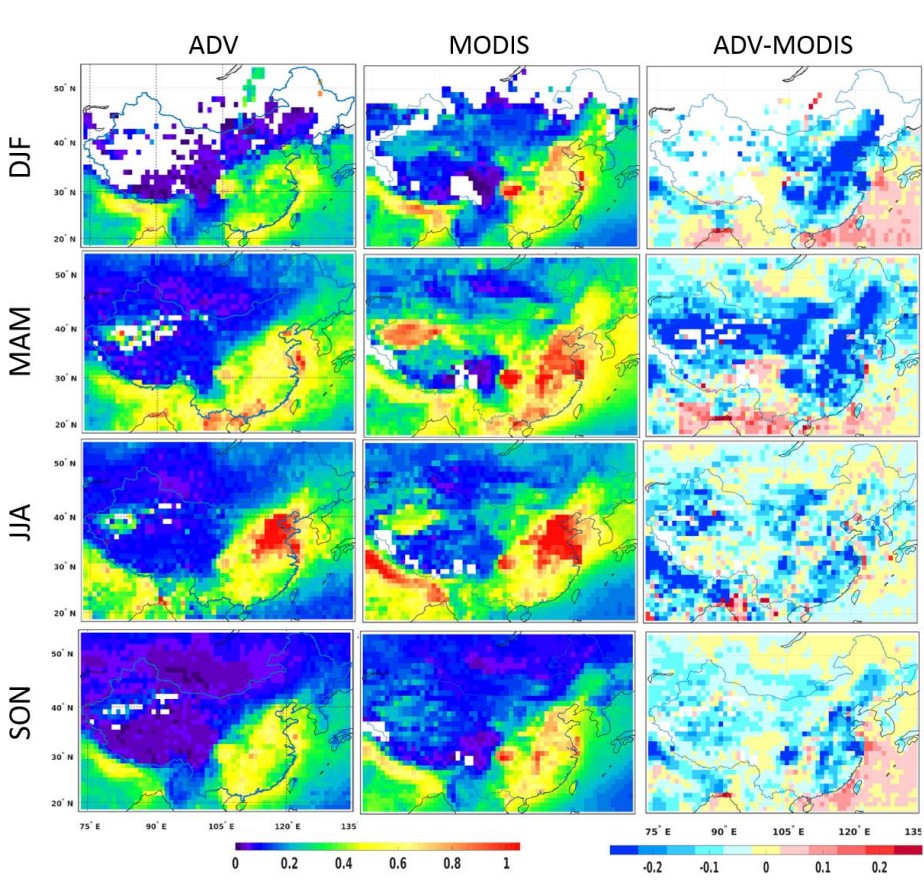

**Figure 10. Seasonal AOD maps aggregated over the years 2000-2011 (top to bottom: DJF (winter), MAM (spring), JJA (summer) and SON (autumn)). Left: ATSR ADV v2.31 (adapted from de Leeuw et al., 2018, Fig. 11); middle MODIS/Terra C6.1 merged DBDT; right: difference maps ADV-MODIS. The AOD and difference scales are plotted at the bottom. Pixels for which no value was retrieved are plotted in white.**

5 **5.1 Seasonal variation by region for the period 2000-2011.**

AOD seasonal time series for China, SE China and each of the 10 regions over China, selected as described in Sect. 2, are shown in Fig. 11, for both

ADV and MODIS. The data shown in Fig. 11 are averages over the three months in each season, and over the years 2000-2011, i.e. the overlapping



period for ATSR and MODIS/Terra. These time series illustrate the overall behavior, which emerged from the seasonal AOD maps in Fig. 10, i.e. a clear seasonal variation of the AOD over all regions, which is similar for both ADV and MODIS but with MODIS AOD somewhat higher than that from ADV. For all regions, the AOD is lowest in the winter, except for China and for region 6 (Sichuan/Chongqing), where the minimum occurs in autumn. For all regions, the AOD is highest in spring, except for regions 1 and 5 where the maximum AOD is observed in the summer. In region 2, the AOD is similar in spring and summer. The difference in the seasonal variation of the AOD between regions 1 (NCP) and 5 and those further south in region 2 (YRD) and region 7 (PRD), which are all very large urban areas with a high population and large industrial development, is likely due to the different climatological zones. The NCP is situated in a temperate monsoon climate region, the YRD in a subtropical monsoon climate region and region 7 combines regions with a sub-tropical and a tropical monsoon climate, with strong differences in rain-season trends, i.e. precipitation and number of rain days (Song et al., 2011; Kourtidis et al., 2015; Stathopoulos et al., 2017). The East-Asian summer monsoon (EASM) and associated rain patterns over east China (Song et al., 2011) progresses from the south in April to the YRD in the early summer and reaches BTH in July. When the monsoon period ends in August, the rain belt moves back to the south. Precipitation obviously affects the AOD due to wash-out of the aerosol particles, but on the other hand on warm days with high relative humidity the aerosol particles swell and thus small (<100 nm) aerosol particles grow into the optically active size range. As a result, the particle size distribution shifts to larger particles and the aerosol scattering and associated AOD increase (Bian et al., 2014; Zhang et al., 2015). In region 10, in the NE of China with a cooler climate where the EASM does not reach, the AOD maximum occurs in spring.




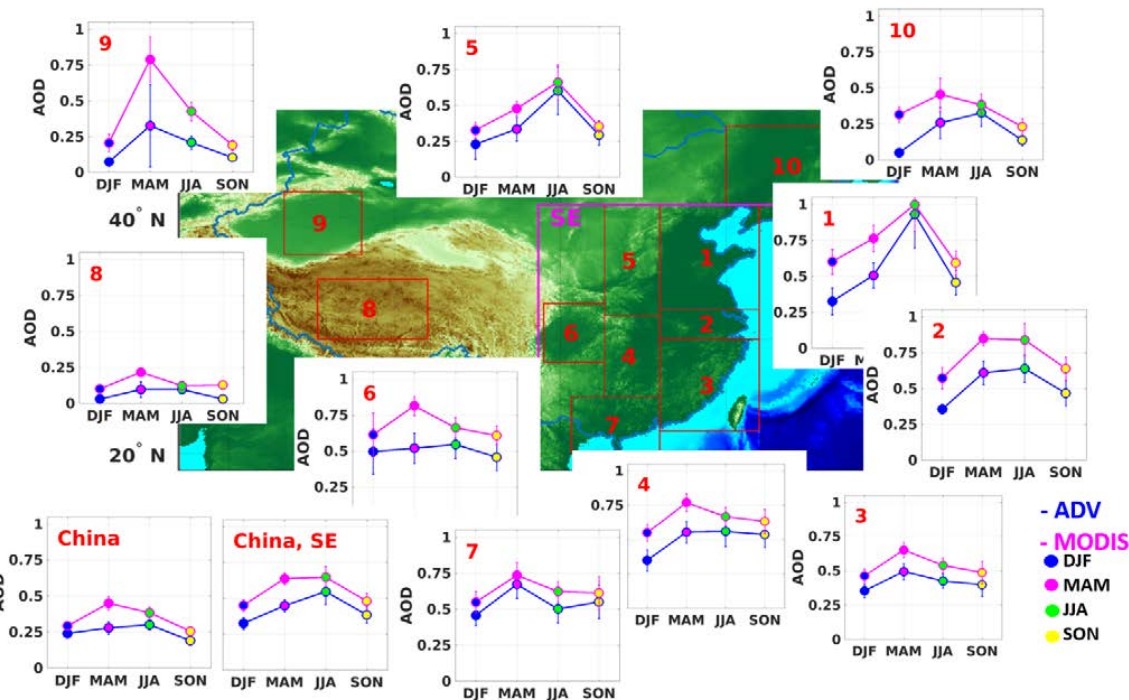

**Figure 11. AOD seasonal time series averaged over the period 2000-2011 for ADV and MODIS (see legend for explanation) for China, SE China and the 10 regions selected as discussed in Sect. 2. Error bars shown on each seasonal data point are one standard deviation.**

5   Another factor influencing the seasonal variation of the AOD is the dust emitted from the deserts with the highest intensity in spring and summer (cf. Proestakis et al., 2018). The largest dust sources in China are the Taklamakan desert and the Gobi deserts. Due to differences in topography, elevation, thermal conditions and atmospheric circulation, the GD has a much more important role than TD in contributing to the dust concentrations in East Asia than the TD (Chen S. et al., 2017). Figure 12, reproduced from Proestakis et al. (2018) , who describe in detail how these products were obtained, shows seasonal maps of Dust AOD (DAOD) at 532 nm, based on CALIOP (Cloud Aerosol Lidar with Orthogonal Polarization ; Winker et al., 2009) observations

10   between 01/2007 and 12/2015. These maps clearly illustrate the effect of the dust generated over the TD, with very high DAOD in the spring (up to about 0.7) and also in the summer, and much lower in the autumn and winter (about 0.2). In contrast, there is no clear dust signal over the northern part of the





Gobi desert where surface dust concentrations are high (cf. Chen S. et al., 2017), neither in the CALIOP DAOD maps in Fig. 12, nor in the ADV and MODIS AOD maps in Figs. 7 and 10. In these satellite observations the dust appears to be confined to south from 40° N.

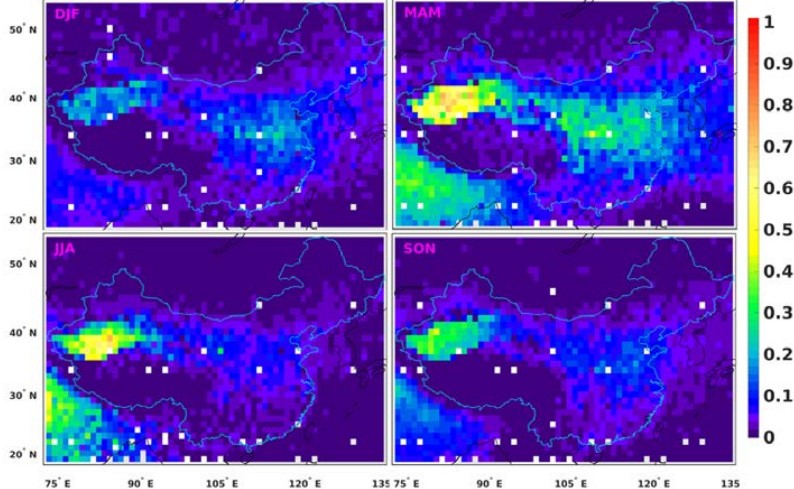

**Figure 12. Spatial distribution of the seasonal mean dust AOD, as determined from CALIPSO observations, aggregated over the period January 2007- December 2015. Adopted from Proestakis et al. (2018). Chinese borders indicated by the blue line.**

A noticeable feature is the distinct eastward pathway of dust aerosol transport, although the observed features strongly vary with season. The eastward dust aerosol pathway extends from the Taklamakan Desert over central China (Kuhlmann and Quaas, 2010), with DAOD values of up to 0.3 in the spring and much smaller in other seasons (0.1), towards the Yellow Sea and the Pacific Ocean (Uno et al., 2009). This dust aerosol Trans-Pacific belt extends over central China between 30° N and 45° N, contributing with dust aerosols up to 50% to the total aerosol load of the densely populated Beijing, Hebei, Tianjin and Shandong provinces (Proestakis et al., 2018). However, very low DAOD values are observed to the south of about 30° N throughout the year, i.e. south of the Yangtze basin, indicating the very low dust aerosol transport to the south of the observed dust aerosol Trans-Pacific belt. The YRD is also the area where the seasonal maximum shifts between spring and summer and North of the YRD are regions 1 and 5 with summer AOD maxima, as described above. Clearly, in spite of the relatively high DAOD over the TD in the summer and presumed sources over the GD, there appears to be little eastward transport and DAOD is not responsible for the high summer AOD with DAOD over the NCP of the order of 0.1. Another candidate for causing the high AOD in summer might be agricultural fires during the summer harvest period in June in the NCP (Zhang et al., 2018) in addition to the mechanism proposed above in reaction to the migration of the EASM.



## 5.2 Long-term AOD variations and their effects on the AOD seasonality: ATSR ADV (1995-2011) and MODIS (2000-2017)

The AOD seasonal time series presented in Fig. 11 are averages over 11 years, during which changes have occurred under the influence of variable meteorological conditions, and interannual variations in large-scale and synoptic situations, as well as economic development and emission control regulations. Hence, the seasonal AOD variations as shown in Fig. 11 may have been different in some years and/or may have evolved over the study period. The influence of the temporal variations on the seasonal bahaviour of the AOD are investigated from the long-term AOD time series for each of the seasons over the selected regions, presented in Fig. 13 for both ADV and MODIS. For clarity of presentation, the time series for these instruments are plotted separately. The temporal variations in the ADV and MODIS data are generally similar. In the following, we will discuss only some of the more significant features such as the different temporal evolution between different seasons in regions 1, 7 and 9 and the change in seasonality in region 2. The ADV and MODIS seasonal AOD tendencies for the overlapping period are discussed below in Sect. 5.3. The AOD temporal evolution and tendencies for 1995-2017 will be discussed in a companion paper (Part II).





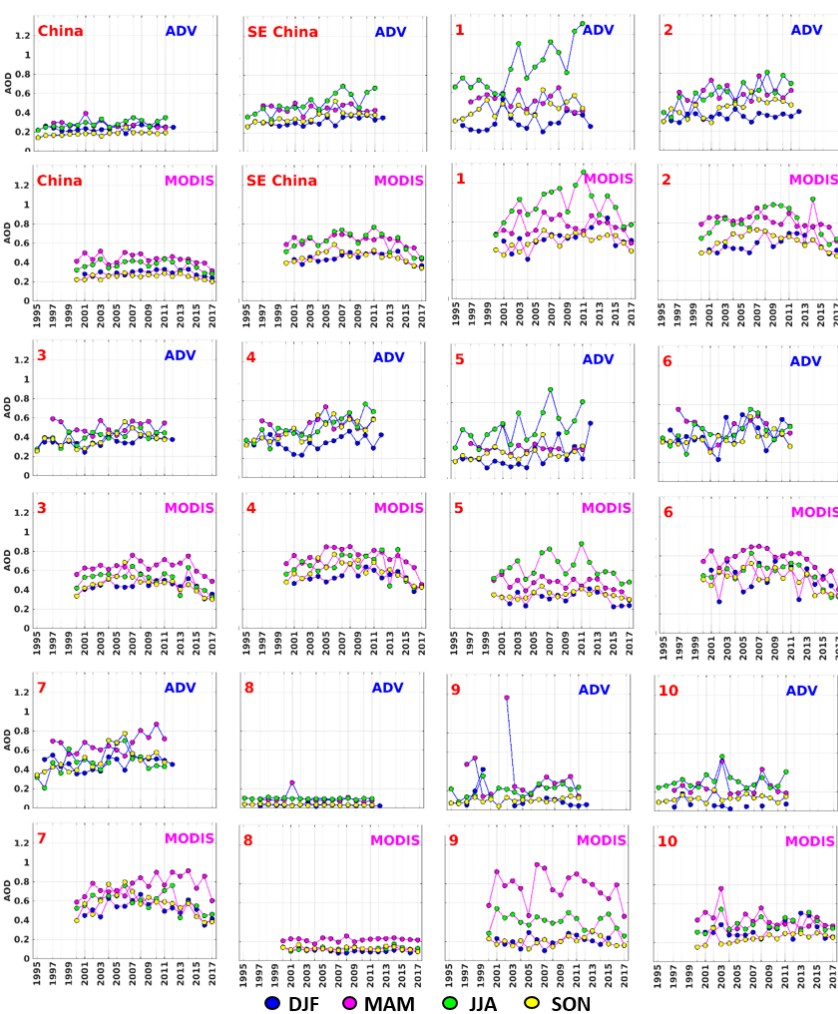

**Figure 13. Time series for seasonally averaged AOD over China, SE China and each of the 10 regions (red numbers or names), for ATSR (1995-2011, blue lines) and MODIS C6.1 DBDT (2000-2017, purple lines) for winter ( DJF, blue dots), spring (MAM, green dots), summer (JJA, purple dots), and autumn (SON, yellow dots).**





The time series in Figure 13 confirm the observations from Figs. 10 and 11 as regards the seasonal behavior with usually a higher AOD in spring and summer than in autumn and winter. However, these time series also show that the long-term temporal behavior may vary between seasons. This is clearly illustrated by, e.g., the data for region 1 where a very strong increase is observed in the summer AOD after 2001, in both the ADV and MODIS data. Before 2001, the summer AOD is higher than in any other season, by about a factor of 1.5 (MODIS data) to 4 (ADV data) as compared to the winter. This large difference between ADV and MODIS is due to an underestimation by ADV in the winter. After 2001, the summer AOD strongly increases whereas for other seasons the AOD remains similar or increases by a small fraction (e.g. in the autumn). After 2011 the MODIS data show the strong decrease of the summer AOD in response to the implementation of emission control policy which appears to most effective for the summer AOD. The much stronger increase of the AOD in summer than in other seasons is also observed in region 5. Because the AOD in region 1 is higher than anywhere else (see Fig. 10) and because regions 1 and 5 together are quite large, the effect of the high summer AOD is also reflected in the AOD aggregated over all SE China and even over all China. As expected, over China and SE China the spring AOD is much closer to that in summer because in all other regions than 1 and 5 the AOD reaches its maximum value in spring, albeit that this maximum value is smaller than in regions 1 and 5 (cf. Figs. 10 and 11).

Another noticeable separation between the AOD behavior in different seasons is observed for region 7, i.e. in the south of China including the provinces Guangxi and Guangdong. Until about 2006, there is little difference between the AOD in different seasons, with winter and summer on the low side and spring and autumn somewhat higher. However, after 2006, the spring AOD is clearly higher and increasing until about 2014, while in the other seasons the AOD appears to decrease. This decrease may be due to the restriction of $SO_2$ emissions resulting in a strong decrease of $SO_2$ concentrations (cf. Koukouli et al., 2016).

A clear separation between seasons is also observed in the MODIS data over region 9 (where ADV does not provide reliable AOD and therefore is not discussed). Region 9 includes the Taklamakan desert and as shown in Fig 12, the dust AOD is highest in the spring, while also in the summer the DAOD is high. In autumn and winter there is very little dust generation and the DAOD are much lower than in spring and summer. This is also reflected in Fig. 13 and apart from interannual variations, the data also suggest a decrease in the AOD in summer, with a stronger decrease in AOD in the spring between 2011 and 2017. These long term variations will be further discussed in Part II.

An area for which the seasonality shifts between spring and summer is region 2. The region 2 time series in Fig. 13 shows that indeed the spring and summer AOD are similar, as shown in Fig. 11, with a slightly higher value in spring before 2007, and a clearly higher value in summer in the years 2008-2012. Hence the seasonal behavior in region 2 would depend on the period considered. After 2012, there is no clear seasonal behavior for this region.



### 5.3 Comparison between ATSR ADV and MODIS seasonal AOD tendencies.

To compare the seasonal year-to-year behavior of AOD retrieved with ATSR and MODIS, the AOD tendencies for the overlapping period (2000-2011) have been estimated by fitting the time series with linear regression lines (Figs. S1-S4). The linear fit for AOD seasonal tendencies for the overlapping period for ADV and MODIS is shown in Figs. S1-S4, the corresponding statistics are shown in Table S2.

For the overlapping period, positive AOD tendencies have been observed with both instruments over China for all seasons, except for spring, when the AOD tendency was close to zero or slightly negative. In winter, the ADV-retrieved AOD shows the strong increase (between 1.31 and 1.51 per decade) in regions 4 to 7, which represent the south and east of SE China. Interestingly, along the east coast, the AOD tendency increases in winter from north to south, as shown with ADV. MODIS shows a strong (near 0.16) AOD increase in winter in regions 1 and 2. In spring, the AOD tendencies are very low for both instruments, showing an increase in the MODIS AOD and a decrease for ADV. The highest AOD increase was observed in region 7 for

both ADV and MODIS (0.181 and 0.171 per decade, respectively). In summer, the strong AOD increase is observed in region 1 for both ADV and MODIS (0.503 and 0.422 per decade, respectively). The positive AOD tendencies were higher in SE China, reaching 0.168 and 0.154 for ADV and MODIS respectively. In autumn, the AOD tendencies were smaller for both ADV and MODIS and agreed in sign for most of the selected regions, except for region 10. Note that the AOD tendencies were statistically significant for regions 1, 2, 4 and over SE China for MODIS only.

The AOD tendencies for the overlapping period derived from MODIS are plotted in Fig. 14 versus those derived from ATSR. This scatterplot includes

tendencies for yearly and seasonal AOD aggregates (dots, see legend for colors) for China, SE China and for each of the 10 selected regions. The confidence for linear fits (p-value) is indicated by the colored (with respect to p-value range) circles around each symbol. The areas where both instruments show similar tendency are colored with light red (both positive) and light blue (both negative) background. The same plot, but with symbols replaced with region numbers is presents in Fig. 14, right.

Most of the ADV and MODIS AOD tendencies for corresponded periods are located in the colored (red and blue) areas, which confirm that ADV and

MODIS show similar in sign AOD tendencies during the overlapping period. The grouping of the tendency points along 1:1 line (line is not shown here) show that in absolute numbers the AOD tendencies are similar for ADV and MODIS.

However, seasonal differences exist in agreement between the ADV and MODIS AOD tendencies. The AOD tendencies derived from the two instruments are in good agreement in summer, autumn and annually (R is 0.87, 0.77, and 0.88, respectively). In winter and spring, the correlation coefficient is smaller (0.41).

Thus, the consistency between ATSR and MODIS as regards the AOD tendencies in the overlapping period is rather strong in the summer, the autumn and for the yearly average, while in the winter and spring, when there is a difference in coverage between the two instruments (Table S1), the agreement in the AOD tendencies is lower.




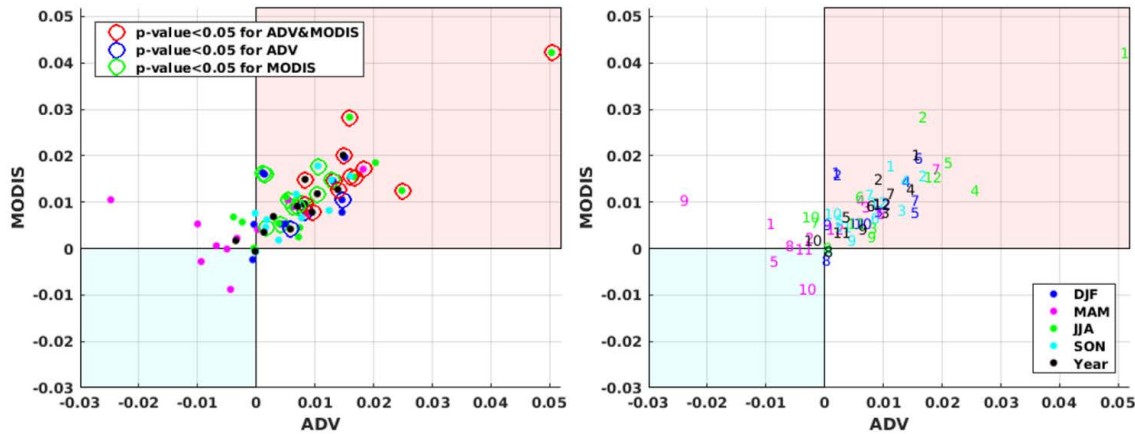

**Figure 14. Scatterplot of the 2000-2011 yearly tendencies (left) derived from MODIS C6.1 DBDT vs. those derived from ATSR ADV v2.31, for China, SE China and 10 selected areas (as specified on the right )for the yearly (black dots) and seasonally averaged AOD (coloured dots, see legend). Coloured circles indicate if p-value<0.05 for both ADV and MODIS (red), only ADV (blue) and only MODIS (green).**

**6. Summary and Conclusions**

The current manuscript is the extension of the study by de Leeuw et al. (2018), where ATSR-retrieved AOD using ADV v2.31 for 1995-2011 and the MODIS/Terra C6 DBDT merged AOD product for 2000-2015 were explored. In the current paper, the MODIS/Terra C6 DBDT merged AOD product has been replaced with the recently released collection C6.1 and extended to also include 2016 and 2017. The AOD annual anomaly maps are shown and discussed; the analysis of the seasonal variability has been extended to 10 selected regions; the AOD tendencies for the overlapping period (2000-2011)

for both ADV and MODIS are presented and compared.

The main results and following conclusions are summarized below.

     - The comparison with AERONET shows the improved performance of C6.1. Thus, the correlation coefficient increased from 0.9 in C6 to 0.92 in C6.1; bias slightly decreased from 0.007 to 0.006. The AOD spatial coverage changed slightly, i.e. in C6.1 it is somewhat lower over bright surfaces and somewhat higher over other areas, as concluded from the increased number of validation points.

- AOD validation with AERONET shows that the validation results depend on the sampling. If the sampling includes all available collocations with AERONET (1132 and 4964 points for ADV and MODIS, respectively), the validation statistics are slightly better for MODIS. The bias in both data



sets is similar but with opposite sign (0.06 for MODIS and -0.07 for ADV). However, for collocated points, when ATSR and MODIS over passes are within +-90 min and AERONET data exist for validation, the correlation coefficient is closer between ADV and MODIS (0.92 and 0.93, respectively) for all collocated points. However, ADV performs better than MODIS in autumn, while MODIS performs slightly better in spring and summer. In winter, both ADV and MODIS underestimated AERONET AOD.

-The AOD interannual variability over China was presented based on annual anomaly maps (with respect to the 2000-2011 averages). During the period 1995-2006, AOD was increasing in the SE of China, while no significant changes in AOD have been observed in the west and in the north. Between 2006 and 2011, AOD was not changing much, showing minor minima in 2008-2009. From 2011 onward, AOD is observed to decrease in the SE of China.

- Both ADV and MODIS show similar seasonal behavior, with spring AOD maxima in the south and shifting from spring to summer along the
eastern coast in the direction to the north.

- Similar patterns are shown in year-to-year differences for ASTR ADV and MODIS AOD. For the overlapping period, positive AOD tendencies have been observed with both instruments over China for all seasons, except for spring, when the AOD tendency was close to zero or slightly negative. More pronounced changes in AOD have been confirmed for SE China. AOD was changing faster in spring and autumn, compared to other seasons.

- The consistency between ATSR and MODIS as regards the AOD tendencies in the overlapping period is rather strong in the summer, the
autumn and for the yearly average, while in the winter and spring, when there is a difference in coverage between the two instruments, the agreement in AOD tendency is lower.

The overall conclusion is that both ATSR ADV and MODIS individually show similar spatial and temporal AOD patterns over China. That conclusion is used as a main starting point in Part II, where the combined long-term AOD time series over China and selected areas will be introduced for the period of 1995-2017. In Part II, AOD tendencies in the combined time series will be estimated for the periods associated with changes in air pollution control
policies in China.

**Data availability**

The ATSR data used in this manuscript are publicly available (after registration a password will be issued) at: http://www.icare.univ-lille1.fr/. MODIS data are publicly available at: https://ladsweb.modaps.eosdis.nasa.gov/ AERONET data are available at AERONET: https://aeronet.gsfc.nasa.gov/

**Acknowledgements**:




Work presented in this contribution was undertaken as part of the MarcoPolo project supported by the EU, FP7 SPACE Grant agreement no. 606953 and as part of the Globemission project ESA-ESRIN Data Users Element (DUE), project AO/1-6721/11/I-NB, and contributes to the ESA/MOST DRAGON4 program. The ATSR algorithm (ADV/ASV) used in this work is improved with support from ESA as part of the Climate Change Initiative (CCI) project Aerosol_cci (ESA-ESRIN projects AO/1-6207/09/I-LG and ESRIN/400010987 4/14/1-NB). Further support was received from the

Centre of Excellence in Atmospheric Science funded by the Finnish Academy of Sciences Excellence (project no. 272041). Many thanks are expressed to NASA Goddard Space Flight Center (GSFC) Level 1 and Atmosphere Archive and Distribution System (LAADS) (http://ladsweb.nascom.nasa.gov) for making available the L3 MODIS/Terra C6.1 and C6 aerosol data. The AERONET team is acknowledged for establishing and maintaining the AERONET sites used in this study.

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
