# Peer review of "Spatial and seasonal variations of aerosols over China from two decades of multi-satellite observations. Part I: ATSR (1995-2011) and MODIS C6.1 (2000-2017)"

_Atmospheric Chemistry and Physics, 2018_

## Referee Comment (RC1) · Anonymous Referee #3 · 20 Apr 2018

The authors analyzed the AOD obtained from ATSR dual view observation jointly with MODIS DTDB results over China for the period since 1995. With such a long time series AOD data from two satellite sensors, the seasonal and inter-annual as well as spatial variation of 2 dataset are compared in details. The results would be useful for researchers who are interesting in the historical aerosol information over East Asia region and thus it is worth the publication. The reviewer's concerns are mainly as below: (1) Over Sichuan basin, its higher AOD has been recognized for a long time. Several ground-based measurement studies also partly confirmed this high AOD region. From

Fig.7, this feature is a major characteristic for MODIS results, while it is not clear for ATSR results. Are there validations or explanation on this underestimation? (2) Over Inner-Mongolia, there are large areas of desert and arid regions, about 1/3 of Takla-makan desert in size. It's somewhat strange why ATSR results show nearly none of higher AOD values over this region. In comparison, MODIS results are probably more reasonable. Noticeably, in Fig 3(a), the "red" color region (difference large than ∼0.2) is also around this area. (3) As shown in Fig 9, the difference between two dataset depends on geolocations and seasons (Fig.11). Then, how does these two dataset (MODIS & ATSR) could be merged or used together in a continuous basis for a future analyses of long period variation, e.g. 1995-present, as suggested by authors in the summary part.

---

## Referee Comment (RC2) · A. M. Sayer (Referee) · 15 May 2018

Summary:

I am writing this review under my own name (Andrew Sayer) as I have previously discussed this research with the authors, and am on the team responsible for the MODIS aerosol data products being used in the study. I also reviewed the paper de Leeuw et al (2018), which is in some sense a predecessor to this study, and have been invited to review the follow up Part II to this paper also by Sogacheva et al and also currently in

[Figure]

ACPD. I feel I am able to provide an impartial review, but am signing the review in the interests of transparency.

The goal of this paper is to look at spatial and temporal (seasonal/interannual) variations of AOD over China. This is accomplished mainly by using two satellite data sets: the ADV algorithm applied to the combined ATSR2/AATSR record (1995-2012), and the combined Deep Blue/Dark Target algorithms applied to the MODIS Terra record (2000 onwards) from the latest Collection 6.1. CALIOP data are also used briefly to look at seasonality of desert dust. AERONET data are also used for validate the MODIS and ATSR retrievals where possible – the available sites are largely confined to eastern China. This paper sets up the later analysis in Part II, which is more focused on combining the time series to look at trends, although there's a bit of overlap in that trends (or at least 'tendencies' in the authors' terminology) are discussed in this paper as well. As a result I felt it best to review this Part I first. Aerosol remote sensing over China can be more difficult than over many other parts of the world, there are less publicly-available validation data than some other areas, and there are suggestions from the literature that historical trends in AOD in some areas may be changing (i.e. they're not linear and start/end dates of analyses are important). So the research question behind the authors' series of papers is relevant. There is overlap between this paper and de Leeuw et al (2018) but the analysis here is expanded and updated, and I think there is sufficient new stuff here that the overlap does not preclude publication.

My overall recommendation is for major revisions and re-review. Overall it is a good paper but I think there's a gap in establishing the reliability of the ATSR2 portion of the record, and the premise of the analysis rests of being able to treat ATSR2 and AATSR as one consistent long-term record. This has not really been assessed by the paper. In my specific comments below I have a few suggestions how that could be done.

The quality of language is overall good and any issues can probably be dealt with by Copernicus' copy-editing and typesetting process. Therefore my review mainly concentrates on technical abstracts. Here, PXLY refers to page X, line Y.

Specific comments:

Abstract: I would condense this into one paragraph if possible and shorten it to highlight the main findings. For example I'd delete the paragraph comparing validation statistics for MODIS C6 and C6.1 as probably not the most important thing (see also later comments on that). I'd also consider shortening or removing the final paragraphs about AOD tendencies over the period, since that's the focus of part II. In my mind it is enough here to talk about the overall validation results and spatial/seasonal patterns. The sentence on P2L5-6 which briefly summarises the interannual variability is useful.

P5L9-10: I would assume that "China" here is defined as any 1x1 degree grid cell with retrievals over land which crosses or is contained by the borders shown in Figure 2. It would be good to make this explicit.

Figure 2: It is hard to see the regional boxes and borders over the top of the AOD map, because there are a lot of colours and some lines are thin. I suggest plotting the national borders in black rather than blue, to make them stand out, as the blue colour used is similar to that coded in the colour bar for common values of AOD shown in the map. Subregion boundaries could also do with having thicker lines and being in a strong contrasting colour. For example although the colour scale tops out at red, red may be a suitable choice as it doesn't look like any of the grid cells reach this value. Alternatively (and probably even better), the authors could abandon the rainbow colour scale used in this Figure and some others in the paper. Rainbow scales can present difficulties for colour blind readers, do not always display or print consistently, and draw the eye to specific points in the range while smoothing variations in other parts. See for example https://www.climate-lab-book.ac.uk/2014/end-of-the-rainbow/ or https://eagereyes.org/basics/rainbow-color-map for some good discussion of why this is, and https://personal.sron.nl/~pault/ for some good alternative palettes to use.

Section 3.1: somewhere in here I would cite Sayer et al (JGR 2014) as the reference for the MODIS merged Deep Blue/Dark Target data set used. This describes the algorithm, shows some examples, and provides validation. The paper is about Collection 6 but the merging logic is unchanged for Collection 6.1 (C61) so it still stands as the best reference. We don't have a paper specific to C61 at present. This study is cited elsewhere in the manuscript already, but I think it would be good to add it here too.

It would be useful to be clear about exactly which products (and which SDS names within them) were used. For example P6L11 says "L3" data – there are several different L3 aggregation time scales and it isn't clear if the daily product (MOD08_D3) was used as a basis and aggregated to a monthly scale, or if the monthly product (MOD08_M3) was used directly. Plus, presumably the later validation exercise used the L2 product (MOD04_L2).

P6L20-23: Here the authors link to a summary document about C61 about cloud detection. This is not quite satisfactory since that's not necessarily a stable url and it isn't a document with a defined author or contact point. I reached out to the MODIS calibration and cloud teams and they suggest replacing with a citation to Moeller et al (SPIE, 2017): http://spie.org/Publications/Proceedings/Paper/10.1117/12.2274340. That paper contains more examples of the effect of the crosstalk problem and the Wilson et al (2017) algorithm to fix it on the cloud detection time series. So I'd just cite the Moeller and Wilson papers here instead of linking to the C61 pdf from the MODIS Atmospheres site.

P7L5-6: Here the authors reference by url one of my review comments to de Leeuw et al (2018) on ACP. I don't necessarily think the sentence is needed since the relevant info is more or less shown in their Figure 3. But if the authors want to keep it I would note that ACPD review comments (to which the figure linked was attached) are citable, and this one was doi:10.5194/acp-2017-838-RC1, so that's a better way than giving the url.

Section 3, general (and my main substantive comment): One point that is glossed over in the validation and later discussion is the fact that almost all the available validation

data for ADV here are for the AATSR portion of the record (2002-2012), not the ATSR-2 portion (1995-2002). The instrument characteristics are very similar so one would expect similar performance. The main potential reason for difference would be offsets in the sensors' absolute calibration, which remains a thorny issue for all satellite AOD data sets. On the one hand there what the authors can do here is limited, because without validation data covering both the ATSR-2 and AATSR missions it's hard to say whether the error characteristics of the two instruments are similar or not, and it is definitely not the authors' fault that this is not available. But on the other hand, since the point of this paper and Part II is to look at a long time series, it is unfortunate that the first 5 years of the period (1995-2000) before MODIS, which are one big advantage of the ATSRs compared to many other sensors, are a period where the data set can't effectively be validated. This is doubly unfortunate since trend analyses are particularly sensitive to values at start and end points.

I think it would make the authors' case stronger if they could demonstrate the reliability of the ATSR2 portion of the record in a more concrete way. From when I used to work on the ATSRs (a different algorithm, ORAC), I recall there is about 1 year of overlap between ATSR2 and AATSR (during 2002-2003) when the two instruments were flying on the same orbit track spaced something like 30 minutes apart. Actually I think there is more than 1 year but my memory is that the ERS2 satellite on which ATSR2 flew had some technical issues with its pointing accuracy after 2003. My point is, I am pretty sure there is 1 year of overlapping data, and potentially more, which could be used here.

So one step would be to take this year of overlapping data, and do an analysis of how consistent the ATSR2 and AATSR retrievals are (in terms of AOD retrieved and also if they make similar choices about when to retrieve, i.e. data coverage). This could be a few extra figures/tables showing joint scatter density histograms or the monthly means in the various regions of interests and things like that.

As a second step, there were a few AERONET sites in China in operation during the

ATSR2 period, and during the ATSR2/AATSR overlap period (2002-2003). There will probably not be too many matches with AERONET during this year, but even if you get only a dozen or so where you have AERONET, ATSR2, and AATSR all together, that lets you say something about whether the retrieval errors are similar or not.

A third option is do this sort of 3-way comparison using ATSR2, AATSR, and MODIS Terra, using the 3-way overlapping time period, which should get more spatial coverage than AERONET. Even though MODIS can't be considered a reference truth, one can still see if the ATSR2 vs. MODIS and AATSR vs. MODIS differences are consistent.

In my view adding analyses along these lines would put the discussions later and in Part II of the manuscript on much firmer ground. Otherwise the study is assuming that ATSR2 can be used to extend the AATSR record back in time, with consistent error characteristics, without directly testing that assumption and assessing quantitatively (via these comparisons) the differences and their implications for long-term analyses. This was also missing from de Leeuw et al (2018) so adding here would further help distinguish this study from that.

One more note on this topic: the stability of the calibration of the ATSR2/AATSR sensors is well established, thanks to on board and vicarious calibration techniques. It is not that which I am concerned about so much in terms of the combined ATSR2/AATSR record. Rather, it is small differences in the absolute calibration (on-board only monitors stability i.e. relative calibration and absolute is based on pre-launch measurements), and small differences in band spectral response functions, which might lead to offsets in the retrieved AOD between the sensors, and thus affect apparent AOD trends/tendencies.

P9L3-9: The differences between C6 and C6.1 performance seem to be sufficiently small to me that I don't think the statement that MODIS C6.1 is "slightly better" is warranted. My hunch is that the differences are not statistically significant given the limited sample sizes. If I were writing this I would say that C6.1 has about 5 percent

more matchups with AERONET but performance is about the same. Also, the sentence about bias decreasing "from 0.007 to 0.06" must be incorrect because 0.06 is larger than 0.007. Figure 4 suggests 0.06 for C.61 (and does not show C6) but the Abstract says 0.007 and 0.006. So something is inconsistent here – I wonder if a decimal place has been added or subtracted in one of the cases. My guess is that it should read 0.07 and 0.06.

Section 3.3, general: I like the various breakdowns and subsets of the data. One other way I have found to be useful is to subset the satellite data by some proxy for aerosol "type" using AERONET data. I tend to classify conditions as "background" (AERONET AOD less than some number like 0.2), "fine-dominated" (AOD larger than 0.2 and AE larger than some value like 1), or "coarse-dominated" (AOD larger than 0.2 and AE smaller than 1). Then we can see if biases in the satellite retrievals are associated with particular aerosol conditions. It might be that dust is fine but smoke is biased, for example. Similar splits can be done with surface classifications. This may be instructive when later looking at regional comparisons and understanding why the data sets are different and which may be more reliable in a given situation. I am not saying this sort of analysis absolutely must be added to the paper, but it might be useful for some of the later discussions and when attempting to merge the MODIS and ATSR data sets to estimate trends in Part II.

P11L1: Here it says a 90 minute temporal window is used. Previously on P9L4, the text said 1 hour. Which is correct, or were different thresholds used for the separate analyses? This should be clarified.

P13L3: The authors tend to use the term "bias" when comparing any two data sets in the paper and describing one as lower/higher than the other. However "bias" is a loaded word with the implication that whatever is described is incorrect by that amount. I think it is appropriate to use the term "bias" when comparing something to AERONET, since AERONET is taken as our ground truth. However when comparing the satellite data sets to one another, as at P13L3, it is better to use the word "offset" instead (i.e. x

is offset from y rather than x is biased against y). I think this more precise terminology will help clarify when we are really talking about something which might be an error, and when we are talking about something that is a difference but the error is uncertain. This would also help a reader who is less familiar with the satellite data and might not realise what typical levels of uncertainty in sun photometer or satellite AOD data sets are.

Figure 7: I would remove this figure because I do not think that annual maps like this are particularly meaningful. Clouds/snow show seasonal dependence, affecting coverage. Aerosol loading and type and surface cover show seasonal dependence, affecting error characteristics. So a set of seasonal maps would be better. These are already provided and discussed later as Figure 10. So I suggest just deleting Figure 7 and the sentences about it and save that discussion for Figure 10 instead. I don't think that having both Figure 7 and Figure 10 is necessary, and Figure 10 is the more informative of the two.

Section 4 and Figures 8 and 9 and discussion: To help focus the manuscript, I would also consider removing these figures and associated discussion (or at least shortening it somewhat). From the outset the goal of this paper is to assess the MODIS and ATSR records, and Part II is meant to be about the trends. So I would try to stick to this focus to make the paper more readable, and also avoid readers of Part II either having to refer back to the trend discussion here or having duplicate material if it is reproduced. Otherwise the subject matter of the two papers is too mixed and you may as well just combine them into one paper. I feel similarly about sections 5.2 and 5.3. Reading through Part II, I feel that this material (which is intended to make the point that the two data sets can be combined, after some correction) feels much more natural if it is moved to an early place in the Part II paper. To me it is more or less part of the method for Part II, not the analysis of Part I. This will also decrease the need for repetition of content between the two papers and allow more space for a comparison between ATSR2 and AATSR during their overlapping year(s) of 2002-2003 as suggested.
I don't think that taking out the interannual variability here would weaken the conclusions or argument for Part I. I understand why the authors initially put this here and that they want to make the analysis more distinct from de Leeuw et al (2018), but I really think the interannual stuff belongs in Part II. Another risk is that a reader reads Part II and sees the tendency/trend calculations here and takes that as the final word, and never goes on to see Part II. So I really feel that switching around this material between the papers is worthwhile for both parts to maintain their focus and readability. If the authors really want to include some interannual stuff here, that's fine, but I would keep it brief and only show a few time series to illustrate the overall magnitudes and interannual variability in ATSR/MODIS without trying to fit linear trends to it. But I don't think that's needed and would rather the space was spent establishing whether ATSR2 and AATSR ADV AOD retrievals can be treated as one seamless AOD record rather than a pair of records from similar instrument.

P31L12-14: as mentioned earlier, I'd instead say that the performance of MODIS C6 and C6.1 are pretty much the same except coverage increased a bit.

---

## Author Comment (AC1) · 20 Jun 2018

We thank the Referee for careful consideration of the manuscript and the positive recommendation. Below we copy the referee's comments in *Italics* and responses are given below each concern.

*The authors analyzed the AOD obtained from ATSR dual view observation jointly with MODIS DTDB results over China for the period since 1995. With such a long time series AOD data from two satellite sensors, the seasonal and inter-annual as well as spatial variation of 2 dataset are compared in details. The results would be useful for researchers who are interesting in the historical aerosol information over East Asia region and thus it is worth the publication. The reviewer's concerns are mainly as below:*

*(1) Over Sichuan basin, its higher AOD has been recognized for a long time. Several ground-based measurement studies also partly confirmed this high AOD region. From Fig.7, this feature is a major characteristic for MODIS results, while it is not clear for ATSR results. Are there validations or explanation on this underestimation?*

Response: Indeed the AOD over Sichuan Province is high! Both in the ATSR and the MODIS retrieval Sichuan AOD is higher than its surroundings. However, ATSR AOD is a bit underestimated and the MODIS AOD is a bit overestimated (as shown in Fig. 4 and also discussed in the text above Fig.7). For the comparison in Fig. 7 we used the same colorscale which shows the MODIS AOD over Sichuan more clear than that from ATSR. By lack of AERONET data we cannot do validation specific for this area. Fig. 10 also shows the high AOD over Sichuan, with similar values for each season, and similar to that over NCP, except for the summer AOD which over NCP is higher than in other seasons.

*(2) Over Inner-Mongolia, there are large areas of desert and arid regions, about 1/3 of Takla- makan desert in size. It's somewhat strange why ATSR results show nearly none of higher AOD values over this region. In comparison, MODIS results are probably more reasonable. Noticeably, in Fig 3(a), the "red" color region (difference large than ~0.2) is also around this area.*

Response: Fig. 3 shows a comparison between MODIS C6.1 and C6, and because we don't have detailed information on the differences between these two MODIS collections from the peer-reviewed literature, we cannot comment on this. The difference between ATSR and MODIS is shown in Figs 7 and 10, and indeed ATSR

does not show the high AOD over the deserts, which we ascribe to the failure of the dual view algorithm to adequately retrieve AOD over bright surface (see, e.g. the text above Fig 7, and references cited there).

*(3) As shown in Fig 9, the difference between two dataset depends on geolocations and seasons (Fig.11). Then, how does these two dataset (MODIS & ATSR) could be merged or used together in a continuous basis for a future analyses of long period variation, e.g. 1995-present, as suggested by authors in the summary part.*

Response: In the last sentence we indicate that the combination of the ATSR and MODIS time series will be discussed in the companion paper, i.e. Part II. Hence this will not be further discussed in the present paper. Part II is available as a discussion paper in ACP, see https://doi.org/10.5194/acp-2018-288.

---

## Author Comment (AC2) · 20 Jun 2018

A. M. Sayer (Referee)

andrew.sayer@nasa.gov

We thank the reviewer Dr. A. M. Sayer for his positive statement and constructive comments. We much appreciate his thoughts and suggestions that reflected in the improvement of this MS. Detailed answers are below. We copy the referee's comments in *Italics* and responses are given below each concern.

*Summary:*

*I am writing this review under my own name (Andrew Sayer) as I have previously dis- cussed this research with the authors, and am on the team responsible for the MODIS aerosol data products being used in the study. I also reviewed the paper de Leeuw et al (2018), which is in some sense a predecessor to this study, and have been invited to review the follow up Part II to this paper also by Sogacheva et al and also currently in ACPD. I feel I am able to provide an impartial review, but am signing the review in the interests of transparency.*

*The goal of this paper is to look at spatial and temporal (seasonal/interannual) variations of AOD over China. This is accomplished mainly by using two satellite data sets: the ADV algorithm applied to the combined ATSR2/AATSR record (1995-2012), and the combined Deep Blue/Dark Target algorithms applied to the MODIS Terra record (2000 onwards) from the latest Collection 6.1. CALIOP data are also used briefly to look at seasonality of desert dust. AERONET data are also used for validate the MODIS and ATSR retrievals where possible – the available sites are largely confined to eastern China. This paper sets up the later analysis in Part II, which is more focused on combining the time series to look at trends, although there's a bit of overlap in that trends (or at least 'tendencies' in the authors' terminology) are discussed in this paper as well. As a result I felt it best to review this Part I first. Aerosol remote sensing over China can be more difficult than over many other parts of the world, there are less publicly- available validation data than some other areas, and there are suggestions from the literature that historical trends in AOD in some areas may be changing (i.e. they're not linear and start/end dates of analyses are important). So the research question behind the authors' series of papers is relevant. There is overlap between this paper and de Leeuw et al (2018) but the analysis here is expanded and updated, and I think there is sufficient new stuff here that the overlap does not preclude publication.*

*My overall recommendation is for major revisions and re-review. Overall it is a good paper but I think there's a gap in establishing the reliability of the ATSR2 portion of the record, and the premise of the analysis rests of being able to treat ATSR2 and AATSR as one consistent long-term record. This has not really been assessed by the paper. In my specific comments below I have a few suggestions how that could be done.*

The comparison between ATSR-2 and AATSR is included as a separate Section. The results of the comparison, to our opinion, prove the possibility to combine the AOD retrieved from two instruments into one data set without the offset corrections. Details are below.

*The quality of language is overall good and any issues can probably be dealt with by Copernicus' copy-editing and typesetting process. Therefore my review mainly concentrates on technical abstracts. Here, PXLY refers to page X, line Y.*

*Specific comments:*

*Abstract: I would condense this into one paragraph if possible and shorten it to high- light the main findings. For example I'd delete the paragraph comparing validation statistics for MODIS C6 and C6.1 as probably not the most important thing (see also later comments on that). I'd also consider shortening or removing the final paragraphs about AOD tendencies over the period, since that's the focus of part II. In my mind it is enough here to talk about the overall validation results and spatial/seasonal patterns. The sentence on P2L5-6 which briefly summarises the interannual variability is useful.*

The Abstract is shortened according to the reviewer suggestions.

*P5L9-10: I would assume that "China" here is defined as any 1x1 degree grid cell with retrievals over land which crosses or is contained by the borders shown in Figure 2. It would be good to make this explicit.*

The text is modified as suggested.

*Figure 2: It is hard to see the regional boxes and borders over the top of the AOD map, because there are a lot of colours and some lines are thin. I suggest plotting the national borders in black rather than blue, to make them stand out, as the blue colour used is similar to that coded in the colour bar for common values of AOD shown in the map. Subregion boundaries could also do with having thicker lines and being in a strong contrasting colour. For example although the colour scale tops out at red, red may be a suitable choice as it doesn't look like any of the grid cells reach this value. Alternatively (and probably even better), the authors could abandon the rainbow colour scale used in this Figure and some others in the paper. Rainbow scales can present difficulties for colour blind readers, do not always display or print consistently, and draw the eye to specific points in the range while smoothing variations in other parts. See for example [https://www.climate-lab-book.ac.uk/2014/end-of-the-r](https://www.climate-lab-book.ac.uk/2014/end-of-the-r)ainbow/ or https://eagereyes.org/basics/rainbow-color-map for some good discussion of why this is, and https://personal.sron.nl/~pault/ for some good alternative palettes to use.*

The colorscale used for all AOD maps is reconsidered. The China border is plotted with black colours, as suggested. The line width for subregions boundaries is adjusted.

*Section 3.1: somewhere in here I would cite Sayer et al (JGR 2014) as the reference for the MODIS merged Deep Blue/Dark Target data set used. This describes the algorithm, shows some examples, and provides validation. The paper is about Collection 6 but the merging logic is unchanged for Collection 6.1 (C61) so it still stands as the best reference. We don't have a paper specific to C61 at present. This study is cited elsewhere in the manuscript already, but I think it would be good to add it here too.*

The citation of the paper by Sayer (JGR, 2014) is added to the current section.

*It would be useful to be clear about exactly which products (and which SDS names within them) were used. For example P6L11 says "L3" data – there are several different L3 aggregation time scales and it isn't clear if the daily product (MOD08_D3) was used as a basis and aggregated to a monthly scale, or if the monthly product (MOD08_M3) was used directly. Plus, presumably the later validation exercise used the L2 product (MOD04_L2).*

The text was modified to clarify the product used:

"In short, L3 (averaged on a grid of 1ox1o) monthly AOD data retrieved from ATSR-2 (1995-2002) and AATSR (2002-2012) (together referred to as ATSR) using ADV version 2.31 (Kolmonen et al., 2016; Sogacheva et al., 2017) and MODIS/Terra AOD C6.1 merged DTDB (L3) monthly data (MOD08_M3, 2000-2017, https://ladsweb.modaps.eosdis.nasa.gov/ ) were used together to cover the period from 1995-2017."

*P6L20-23: Here the authors link to a summary document about C61 about cloud detection. This is not quite satisfactory since that's not necessarily a stable url and it isn't a document with a defined author or contact point. I reached out to the MODIS calibration and cloud teams and they suggest replacing with a citation to Moeller et al (SPIE, 2017): http://spie.org/Publications/Proceedings/Paper/10.1117/12.2274340. That paper contains more examples of the effect of the crosstalk problem and the Wilson et al (2017) algorithm to fix it on the cloud detection time series. So I'd just cite the Moeller and Wilson papers here instead of linking to the C61 pdf from the MODIS Atmospheres site.*

The reference to Moeller et al (SPIE, 2017) is added. Thanks for the suggestion.

*P7L5-6: Here the authors reference by url one of my review comments to de Leeuw et al (2018) on ACP. I don't necessarily think the sentence is needed since the relevant info is more or less shown in their Figure 3. But if the authors want to keep it I would note that ACPD review comments (to which the figure linked was attached) are citable, and this one was doi:10.5194/acp-2017-838-RC1, so that's a better way than giving the url.*

The reference is corrected as a link to doi:10.5194/acp-2017-838-RC1

*Section 3, general (and my main substantive comment): One point that is glossed over in the validation and later discussion is the fact that almost all the available validation data for ADV here are for the AATSR portion of the record (2002-2012), not the ATSR- 2 portion (1995-2002). The instrument characteristics are very similar so one would expect similar performance. The main potential reason for difference would be offsets in the sensors' absolute calibration, which remains a thorny issue for all satellite AOD data sets. On the one hand there what the authors can do here is limited, because without validation data covering both the ATSR-2 and AATSR missions it's hard to say whether the error characteristics of the two instruments are similar or not, and it is definitely not the authors' fault that this is not available. But on the other hand, since the point of this paper and Part II is to look at a long time series, it is unfortunate that the first 5 years of the period (1995-2000) before MODIS, which are one big advantage of the ATSRs compared to many other sensors, are a period where the data set can't effectively be validated. This is doubly unfortunate since trend analyses are particularly sensitive to values at start and end points.*

*I think it would make the authors' case stronger if they could demonstrate the reliability of the ATSR2 portion of the record in a more concrete way. From when I used to work on the ATSRs (a different algorithm, ORAC), I recall there is about 1 year of overlap between ATSR2 and AATSR (during 2002-2003) when the two instruments*

*were flying on the same orbit track spaced something like 30 minutes apart. Actually I think there is more than 1 year but my memory is that the ERS2 satellite on which ATSR2 flew had some technical issues with its pointing accuracy after 2003. My point is, I am pretty sure there is 1 year of overlapping data, and potentially more, which could be used here.*

*So one step would be to take this year of overlapping data, and do an analysis of how consistent the ATSR2 and AATSR retrievals are (in terms of AOD retrieved and also if they make similar choices about when to retrieve, i.e. data coverage). This could be a few extra figures/tables showing joint scatter density histograms or the monthly means in the various regions of interests and things like that.*

*As a second step, there were a few AERONET sites in China in operation during the ATSR2 period, and during the ATSR2/AATSR overlap period (2002-2003). There will probably not be too many matches with AERONET during this year, but even if you get only a dozen or so where you have AERONET, ATSR2, and AATSR all together, that lets you say something about whether the retrieval errors are similar or not.*

*A third option is do this sort of 3-way comparison using ATSR2, AATSR, and MODIS Terra, using the 3-way overlapping time period, which should get more spatial coverage than AERONET. Even though MODIS can't be considered a reference truth, one can still see if the ATSR2 vs. MODIS and AATSR vs. MODIS differences are consistent.*

*In my view adding analyses along these lines would put the discussions later and in Part II of the manuscript on much firmer ground. Otherwise the study is assuming that ATSR2 can be used to extend the AATSR record back in time, with consistent error characteristics, without directly testing that assumption and assessing quantitatively (via these comparisons) the differences and their implications for long-term analyses. This was also missing from de Leeuw et al (2018) so adding here would further help distinguish this study from that.*

*One more note on this topic: the stability of the calibration of the ATSR2/AATSR sensors is well established, thanks to on board and vicarious calibration techniques. It is not that which I am concerned about so much in terms of the combined ATSR2/AATSR record. Rather, it is small differences in the absolute calibration (on-board only monitors stability i.e. relative calibration and absolute is based on pre-launch measurements), and small differences in band spectral response functions, which might lead to offsets in the retrieved AOD between the sensors, and thus affect apparent AOD trends/tendencies.*

We very much appreciate the detailed overview on the ATSR-2 and AATSR technical issues and calibrations. Comparison between the ATSR-2 and ATSR AOD is done by looking at the L2 pixel-by-pixel difference in AOD over China and monthly means comparison and validation results over China and globally.

*P9L3-9: The differences between C6 and C6.1 performance seem to be sufficiently small to me that I don't think the statement that MODIS C6.1 is "slightly better" is warranted.  My hunch is that the differences are not statistically significant given the limited sample sizes. If I were writing this I would say that C6.1 has about 5 percent more matchups with AERONET but performance is about the same. Also, the sentence about bias decreasing "from 0.007 to 0.06" must be incorrect because 0.06 is larger than 0.007. Figure 4 suggests 0.06 for C.61 (and does not show C6) but the Abstract says 0.007 and 0.006. So something is inconsistent here – I wonder if a decimal place has been added or subtracted in one of the cases. My guess is that it should read 0.07 and 0.06.*

The conclusion on the MODIS C6 and C6.1 AOD comparison is modified as you suggested. Numbers for C6 and C6.1 biases are corrected in the Conclusions (not in the Abstract).

*Section 3.3, general: I like the various breakdowns and subsets of the data. One other way I have found to be useful is to subset the satellite data by some proxy for aerosol "type" using AERONET data. I tend to classify conditions as "background" (AERONET AOD less than some number like 0.2), "fine-dominated" (AOD larger than 0.2 and AE larger than some value like 1), or "coarse-dominated" (AOD larger than 0.2 and AE smaller than 1). Then we can see if biases in the satellite retrievals are associated with particular aerosol conditions. It might be that dust is fine but smoke is biased, for example. Similar splits can be done with surface classifications. This may be instructive when later looking at regional comparisons and understanding why the data sets are different and which may be more reliable in a given situation. I am not saying this sort of analysis absolutely must be added to the paper, but it might be useful for some of the later discussions and when attempting to merge the MODIS and ATSR data sets to estimate trends in Part II.*

The validation analysis for different aerosol types, as classified by the AOD and AE from AERONET according your suggestions, is added.

*P11L1: Here it says a 90 minute temporal window is used. Previously on P9L4, the text said 1 hour. Which is correct, or were different thresholds used for the separate analyses? This should be clarified.*

For the validation, the time window of 1h between AERONET and ATSR or MODIS was chosen. For the comparison of the AOD between ATSR and MODIS, the temporal window between those instruments was below 90 min. This is clarified in the text.

*P13L3: The authors tend to use the term "bias" when comparing any two data sets in the paper and describing one as lower/higher than the other. However "bias" is a loaded word with the implication that whatever is described is incorrect by that amount. I think it is appropriate to use the term "bias" when comparing something to AERONET, since AERONET is taken as our ground truth. However when comparing the satellite data sets to one another, as at P13L3, it is better to use the word "offset" instead (i.e. x is offset from y rather than x is biased against y). I think this more precise terminology will help clarify when we are really talking about something which might be an error, and when we are talking about something that is a difference but the error is uncertain. This would also help a reader who is less familiar with the satellite data and might not realise what typical levels of uncertainty in sun photometer or satellite AOD data sets are.*

We fully agree with your suggestion to used "bias" with respect to validation results and used "offset" while comparing AOD from different satellites. The text is corrected accordingly.

*Figure 7: I would remove this figure because I do not think that annual maps like this are particularly meaningful. Clouds/snow show seasonal dependence, affecting coverage. Aerosol loading and type and surface cover show seasonal dependence, affecting error characteristics. So a set of seasonal maps would be better. These are already provided and discussed later as Figure 10. So I suggest just deleting Figure 7 and the sentences about it and save that discussion for Figure 10 instead. I don't think that having both Figure 7 and Figure 10 is necessary, and Figure 10 is the more informative of the two.*

*Section 4 and Figures 8 and 9 and discussion: To help focus the manuscript, I would also consider removing these figures and associated discussion (or at least shortening it somewhat). From the outset the goal of this paper is to assess the MODIS and ATSR records, and Part II is meant to be about the trends. So I would try to stick to this focus to make the paper more readable, and also avoid readers of Part II either having to refer back to the trend discussion here or having duplicate material if it is reproduced. Otherwise the subject matter of the two papers is too mixed and you may as well just combine them into one paper. I feel similarly about sections 5.2 and 5.3. Reading through Part II, I feel that this material (which is intended to make the point that the two*

*data sets can be combined, after some correction) feels much more natural if it is moved to an early place in the Part II paper. To me it is more or less part of the method for Part II, not the analysis of Part I. This will also decrease the need for repetition of content between the two papers and allow more space for a comparison between ATSR2 and AATSR during their overlapping year(s) of 2002-2003 as suggested.*

Sections 4 and 5.2 are deleted. Figure 8 will be moved to Part II and discussed there.

However, we decided to show the ATSR and MODIS AOD time series for summer (other seasons and year are in Supplement) and results for AOD tendencies analysis in Section 5 for the overlapping period 2000-2011 only, since this analysis is relevant for the discussion on the consistency between ATSR and MODIS AOD. We are not looking at the whole time series tendencies, which is the topic of Part II.

*I don't think that taking out the interannual variability here would weaken the conclusions or argument for Part I. I understand why the authors initially put this here and that they want to make the analysis more distinct from de Leeuw et al (2018), but I really think the interannual stuff belongs in Part II. Another risk is that a reader reads Part II and sees the tendency/trend calculations here and takes that as the final word, and never goes on to see Part II. So I really feel that switching around this material between the papers is worthwhile for both parts to maintain their focus and readability. If the authors really want to include some interannual stuff here, that's fine, but I would keep it brief and only show a few time series to illustrate the overall magnitudes and interannual variability in ATSR/MODIS without trying to fit linear trends to it. But I don't think that's needed and would rather the space was spent establishing whether ATSR2 and AATSR ADV AOD retrievals can be treated as one seamless AOD record rather than a pair of records from similar instrument.*

*P31L12-14: as mentioned earlier, I'd instead say that the performance of MODIS C6 and C6.1 are pretty much the same except coverage increased a bit.*

Corrected, as suggested by the reviewer.

---

## Author Comment (AC3) · 25 Jun 2018

This is an additional information to the reply to Anonymous Referee #3.

During the revision, some of the figures and related discussion have been taken away and other figures are added, as suggested by Referee #2.

In the revised version of the manuscript, Fig. 3 (as submitted to ACPD) goes as Fig.6.

Figure 7 and the related discussion have been taken away.

---

## Author Response (AR2)

We thank Dr. Andrew Sayer for positive assessment for changes we made in the manuscript during the first revision process. Below we reply to the comments for the revised version. Referee's comments are in *Italics* and responses are given below each concern.

5 *My main content concerns with the previous version were (1) establishing the consistency between ATSR-2 and AATSR records such that they can be treated as a single data set for trend analyses, and (2) the split of material between this paper and the companion Part II. In this revision the authors have addressed those concerns, by adding substantial new material comparing ATSR-2 and AATSR retrievals, and by reorganising the later sections which had some overlap with Part II. My other comments have also been addressed. I appreciate their efforts and think that the balance between Parts I and II is better now (and also look forward to reviewing the eventual revised Part II). I therefore recommend publication after technical corrections, for the following*
10 *small points:*

*Abstract – I would still like if this could be shorter (one long paragraph) but will leave this up to the Editor as it is more a matter of personal preference.*

15 The abstract has been shortened during the first revision. We think that the main results shorty mentioned in the current version are important and give clear overview for the reader. However, in the current revision, we slightly re-organised paragraphs.

*P7L29: The text should clarify whether 'pixel by pixel' here for this part of the analysis refers to level 2 retrieval pixels (which I think are 0.1 degree) or level 3 1 degree grid cells. (I think either is fine for this type of consistency analysis.)*

In P7L29 (submitted after first revision version) it is already mentioned that pixel-by-pixel comparison has been done for L2 AOD data. "Pixel-by-pixel comparison between the ATSR-2 and AATSR **L2** AOD is shown in Fig. 3 with…..". L2 is defined early.

25 *Figure 5: ATSR-2 and AATSR were on the same orbit track, 30 minutes apart. So I would expect the number of matchups with AERONET across the August-December 2002 period to be very similar, since aerosol conditions and cloud cover should in general be similar. But this figure gives 223 matchups for ATSR-2, and more than double that (536) for AATSR. This suggests to me that something else is going on. Is ATSR-2 systematically only providing retrievals about half the time that AATSR is? If so, why is that? ATSR-2 had a narrow-swath mode for the solar channels due to bandwidth limitations related to the scatterometer also on ERS-2,*
30 *but that was largely operating only over the ocean, so my guess is that this may not be the prime factor, since most AERONET sites are over the continents. Unless it happens that most AERONET sites during this month were near the coast. Or could there be a bug in the code somewhere here? I would appreciate if the authors could check this and add a sentence or two with the explanation for the difference, otherwise other readers may like me be a bit surprised and confused about this as well.*

We did not find any bug in our validation code. However, we have two explanation for the fact that we have less validation points (less matchups) for ATSR-2 compared to AATSR.

First explanation is related to the automatic threshold approach in cloud tests in ADV, which are applied to 512x480 pixels scenes. Once the threshold is not determined, the whole scene is discarded, as over Scandinavia and in central Russia in Fig. 1. This happens more often for ATSR-2 data. Why? Possible explanation is below.

[Figure]

**Figure 1. ADV AOD retrieved from ATSR-2 (left) an AATSR (right) for the 2nd of August 2002.**

Second explanation for less matchups between ATSR-2 and AATSR is shown in Fig. 2, where (in ADV Ver2.31, as well as in previous versions) a clear difference in coverage between ATSR-2 and AATSR also over land exists.

[Figure]

**Figure 2. ADV AOD retrieved from ATSR-2 (left) an AATSR (right) for the 12th of August 2002.**

As shown in Fig.2, a narrow swath from ATSR-2 is retrieved also over land over some areas (Scandinavia and central Russia, in
5    that case), while a full swath is retrieved over land over other areas (e.g., Africa). We tend to think that this relates to the ATSR-2
features rather than to a bug in ADV.

Surprisingly, full swath is retrieved from ATSR-2 over western Mediterranean, while narrow swath is retrieved over eastern
Mediterranean.

We will double check ATSR-2 coverage over land for the next reprocessing. Once it gets more clear, the explanation for the
10    difference in coverage between ATSR-2 and AATSR will be reported.

*Figure 11: Since the x-axis is labelled by season, I don't think this figure needs the symbols coloured differently for each season as
well. It seems a little redundant/distracting, especially since two of the colours overlap with the lines denoting individual data sets
(e.g. blue means ADV, but also means DJF for both ADV and MODIS). I would rather just see ADV in all blue and MODIS in all
15    magenta.*

To our opinion, the additional colouring for the dots helps to read the figure and does not make it heavier. Thus, we decided to keep
Fig.11 as it was in the submitted version.

[revised manuscript text omitted]